# Learning adaptive planning representations with natural language guidance

**Lionel Wong**[1][*]   **Jiayuan Mao**[1][*]   **Pratyusha Sharma**[1][*]   **Zachary S. Siegel**[2]   **Jiahai Feng**[3]
**Noa Korneev**[4]   **Joshua B. Tenenbaum**[1]   **Jacob Andreas**[1]
[1]MIT   [2]Princeton University   [3]UC Berkeley   [4]Microsoft

## Abstract

Effective planning in the real world requires not only world knowledge, but the ability to leverage that knowledge to build the *right representation* of the task at hand. Decades of hierarchical planning techniques have used domain-specific temporal *action abstractions* to support efficient and accurate planning, almost always relying on human priors and domain knowledge to decompose hard tasks into smaller subproblems appropriate for a goal or set of goals. This paper describes *Ada* (Action Domain Acquisition), a framework for automatically constructing task-specific planning representations using task-general background knowledge from language models (LMs). Starting with a general-purpose hierarchical planner and a low-level goal-conditioned policy, Ada **interactively learns a library of planner-compatible high-level action abstractions and low-level controllers adapted to a particular domain of planning tasks**. On two language-guided interactive planning benchmarks (*Mini Minecraft* and *ALFRED Household Tasks*), Ada strongly outperforms other approaches that use LMs for sequential decision-making, offering more accurate plans and better generalization to complex tasks.

## 1 Introduction

People make complex plans over long timescales, flexibly adapting what we *know* about the world in general to govern how we act in specific situations. To make breakfast in the morning, we might convert a broad knowledge of cooking and kitchens into tens of fine-grained motor actions in order to find, crack, and fry a specific egg; to achieve a complex research objective, we might plan a routine over days or weeks that begins with the low-level actions necessary to ride the subway to work. The problem of *adapting general world knowledge to support flexible long-term planning* is one of the unifying challenges of AI. While decades of research have developed representations and algorithms for solving restricted and shorter-term planning problems, generalized and long-horizon planning remains a core, outstanding challenge for essentially all AI paradigms, including classical planning (Erol et al., 1994), reinforcement learning (Sutton et al., 1999), and modern generative AI (Wang et al., 2023a).

How do humans solve this computational challenge? A growing body of work in cognitive science suggests that people come up with *hierarchical, problem-specific representations* of their actions and environment to suit their goals, tailoring how they represent, remember, and reason about the world to plan efficiently for a particular set of tasks (e.g., Ho et al., 2022). In AI, a large body of work has studied *hierarchical planning using domain-specific temporal abstractions*—progressively decomposing high-level goals into sequences abstract actions that eventually bottom out in low-level control. An extensive body of work has explored how to plan using these hierarchical action spaces, including robotic task-and-motion planning (TAMP) systems (Garrett et al., 2021) and hierarchical RL frameworks (Sutton et al., 1999).

However, identifying a set of abstract actions that are relevant and useful for achieving any given set of goals remains the central bottleneck in general. Intuitively, "useful" high-level actions must satisfy many different criteria: they should enable time-efficient high-level planning, correspond feasible low-level action sequences, and compose and generalize to new tasks. Despite efforts to learn high-level actions automatically in both classical planning (Nejati et al., 2006) and RL formulations (Dietterich, 2000), most state-of-the-art robotics and planning systems rely on human expertise to hand-engineer new planning representations for each new domain (Ahn et al., 2022).

---

[*]Asterisk indicates equal contribution. Correspondence to `zyzzyva@mit.edu`. Code for this paper will be released at: https://github.com/CatherineWong/llm-operators

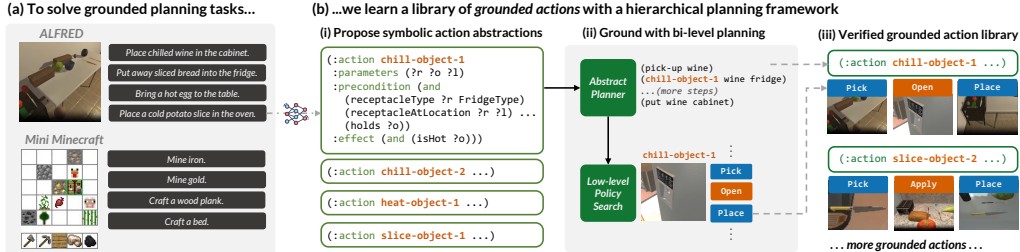

Figure 1: We solve complex planning tasks specified in language and grounded in interactive environments by jointly learning a *library of symbolic high-level action abstractions and modular low-level controllers* associated with each abstraction. Our system leverages background information in language as a prior to *propose useful action abstractions*, then uses a *hierarchical planning framework* to verify and ground them.

In this paper, we introduce *Action Domain Acquisition* (*Ada*), a framework for using background knowledge from language (conveyed via language models) as an initial source of task-relevant domain knowledge. Ada uses language models (LMs) in an interactive planning loop to assemble a *library of composable, hierarchical actions tailored to a given environment and task space*. Each action consists of two components: (1) a *high-level abstraction* represented as a symbolic planning *operator* (Fikes & Nilsson, 1971) that specifies preconditions and action effects as sets of predicates; and (2) a *low-level controller* that can achieve the action's effects by predicting a sequence of low-level actions with a neural network or local search procedure. We study planning in a multitask reinforcement learning framework, in which agents interact with their environments to must solve collections of tasks of varying complexity. Through interaction, Ada incrementally builds a library of actions, ensuring at each step that learned high-level actions compose to produce valid abstract plans and realizable low-level trajectories.

We evaluate Ada (Fig. 1) on two benchmarks, *Mini Minecraft* and *ALFRED* (Shridhar et al., 2020). We compare this approach against three baselines that leverage LMs for sequential decision-making in other ways: to parse linguistic goals into formal specifications that are solved directly by a planner (as in Liu et al. (2023)), to directly predict sequences of high-level subgoals (as in Ahn et al. (2022)), and to predict libraries of actions defined in general imperative code (as in Wang et al. (2023a)). In both domains, we show that Ada learns action abstractions that allow it to solve dramatically more tasks on each benchmark than these baselines, and that these abstractions compose to enable efficient and accurate planning in complex, unseen tasks.

## 2 PROBLEM FORMULATION

We assume access to an environment $\langle \mathcal{X}, \mathcal{U}, \mathcal{T} \rangle$, where $\mathcal{X}$ is the (raw) state space, $\mathcal{U}$ is the (low-level) action space (e.g., robot commands), and $\mathcal{T}$ is a deterministic transition function $\mathcal{T} : \mathcal{X} \times \mathcal{U} \to \mathcal{X}$. We also have a set of features (or "predicates") $\mathcal{P}$ that define an abstract state space $\mathcal{S}$: each abstract state $s \in \mathcal{S}$ is composed of a set of objects and their features. For example, a simple scene that contains bread on a table could be encoded as an abstract state with two objects $A$ and $B$, and atoms $\{bread(A), table(B), on(A, B)\}$. We assume the mapping from environmental states to abstract states $\Phi : \mathcal{X} \to \mathcal{S}$ is given and fixed (though see Migimatsu & Bohg, 2022 for how it might be learned).

In addition to the environment, we have a collection of tasks $t$. Each $t$ is described by a natural language instruction $\ell_t$, corresponding to a goal predicate (which is not directly observed). In this paper, we assume that predicates may be defined in terms of abstract states, i.e., $g_t : \mathcal{S} \to \{T, F\}$. Our goal is to build an agent that, given the initial state $x_0 \in \mathcal{X}$ and the natural language instruction $\ell_t$, can generate a sequence of low-level actions $\{u_1, u_2, \cdots, u_H\} \in \mathcal{U}^H$ such that $g_t(\Phi(x_H))$ is true (where $x_H$ is the terminal state of sequentially applying $\{u_i\}$ on $x_0$). The agent receives reward signal only upon achieving the goal specified by $g_t$.

Given a very large number of interactions, a sufficiently expressive reflex policy could, in principle, learn a policy that maps from low-level states to low-level actions conditioned on the language instruction $\pi(u \mid x; \ell_t)$. However, for very long horizons $H$ and large state spaces (e.g., composed of many objects and compositional goals), such algorithms can be highly inefficient or effectively infeasible. The key idea behind our approach is to use natural language descriptions $\ell_t$ to bootstrap a high-level action space $\mathcal{A}$ over the abstract state space $\mathcal{S}$ to accelerate learning and planning.

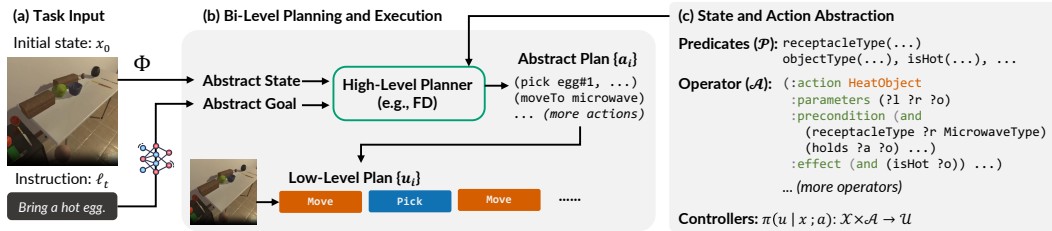

Figure 2: Representation for our (a) task input, (b) the bi-level planning and execution pipeline for inference time, and (c) the abstract state and action representation.

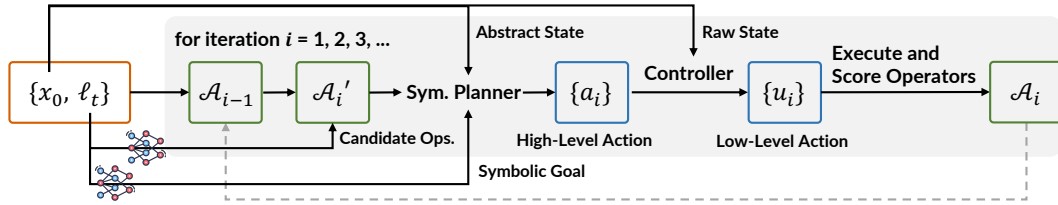

Figure 3: The overall framework. Given task environment states and descriptions, at each iteration, we first propose candidate abstract actions (operators) $\mathcal{A}'_i$, then uses bi-level planning and execution to solve tasks. We add operators to the operator library based on the execution result.

Formally, our approach learns a library of high-level actions (operators) $\mathcal{A}$. As illustrated in Fig. 2b, each $a \in \mathcal{A}$ is a tuple of $\langle name, args, pre, eff, controller \rangle$. *name* is the name of the action, *args* is a list of variables, usually denoted by $?x, ?y, etc.$, *pre* is a precondition formula based on the variables *args* and the features $\mathcal{P}$, and *eff* is the effect, which is also defined in terms of *args* and $\mathcal{P}$. Finally, *controller* : $\mathcal{X} \rightarrow \mathcal{U}$ is a low-level policy associated with the action. The semantics of the preconditions and effects is: for any state $x$ such that $pre(\Phi(x))$, executing *controller* starting in $x$ (for an indefinite number of steps) will yield a state $x'$ such that $eff(\Phi(x'))$ (Lifschitz, 1986). *In this framework, $\mathcal{A}$ defines a partial, abstract world model of the underlying state space.*

As shown in Fig. 2b, given the set of high-level actions and a parse of the instruction $\ell_t$ into a first-order logic formula, we can leverage symbolic planners (e.g., Helmert, 2006) to first compute a high-level plan $\{a_1, \cdots, a_K\} \in \mathcal{A}^K$ that achieves the goal $\ell_t$ symbolically, and then refine the high-level plan into a low-level plan with the action controllers. This bi-level planning approach decomposes long-horizon planning problems into several short-horizon problems. Furthermore, it can also leverage the compositionality of high-level actions $\mathcal{A}$ to generalize to longer plans.

## 3    ACTION ABSTRACTIONS FROM LANGUAGE

As illustrated in Fig. 3, our framework, *Action Domain Acquisition* (*Ada*) learns action abstractions iteratively as it attempts to solve tasks. Our algorithm is given a dataset of tasks and their corresponding language descriptions, the feature set $\mathcal{P}$, and optionally an initial set of high-level action operators $\mathcal{A}_0$. At each iteration $i$, we first use a large language model (LLM) to propose a set of novel high-level action definitions $\mathcal{A}'_i$ based on the features $\mathcal{P}$ and the language goals $\{\ell_t\}$ (Section 3.1). Next, we use a LLM to also translate each language instruction $\ell_t$ into a symbolic goal description $F_t$, and use a bi-level planner to compute a low-level plan to accomplish $\ell_t$ (Section 3.2). Then, based on the planning and execution results, we score each operator in $\mathcal{A}_i$ and add ones to the verified library if they have yielded successful execution results (Section 3.4). To accelerate low-level planning, we simultaneously learn local subgoal-conditioned policies (i.e., the controllers for each operator; Section 3.3). Algorithm 1 summarizes the overall framework.

A core goal of our approach is to adapt the initial action abstractions proposed from an LLM prior into a set of *useful* operators $\mathcal{A}*$ that permit efficient and accurate planning on a dataset of tasks and ideally, that generalize to future tasks. While language provides a key initial prior, our formulation refines and verifies the operator library to adapt to a given planning procedure and environment (similar to other action-learning formulations like Silver et al., 2021). Our formulation ensures not only that the learned operators respect the dynamics of the environment, but also fit their grain of abstraction according to the capacity of the controller, trading off between fast high-level planning and efficient low-level control conditioned on each abstraction.

---

**Algorithm 1** Action Abstraction Learning from Language

---

**Input:** Dataset of tasks and their language descriptions $\{\ell_t\}$
**Input:** Predicate set $\mathcal{P}$
**Input:** Optionally, an initial set of abstract operators $\mathcal{A}_0$, or $\mathcal{A}_0 = \emptyset$
1: Initialize subgoal-conditioned policy $\pi_\theta$.
2: **for** $i = 1, 2, \cdots, M$ **do**
3:    $\mathcal{A}_i \leftarrow \mathcal{A}_{i-1} \cup \text{ProposeOperatorDefinitions}(\mathcal{P}, \{\ell_t\})$            ▷ Section 3.1
4:    **for** each unsolved task $j$: $(x_0^{(j)}, \ell_t^{(j)})$ **do**
5:       $\bar{u} \leftarrow \text{BiLevelPlan}(\mathcal{A}_i, \ell_t^{(j)}, \pi)$                   ▷ Section 3.2
6:       $result^{(j)} \leftarrow \text{Execute}(x_0^{(j)}, \bar{u})$                 ▷ Execute the plan
7:    $\theta \leftarrow \text{UpdateSubgoalPolicy}(\theta, result)$               ▷ Section 3.3
8:    $\mathcal{A}_i \leftarrow \text{ScoreAndFilter}(\mathcal{A}_i, result)$               ▷ Section 3.4
    **return** $\mathcal{A}_M$

---

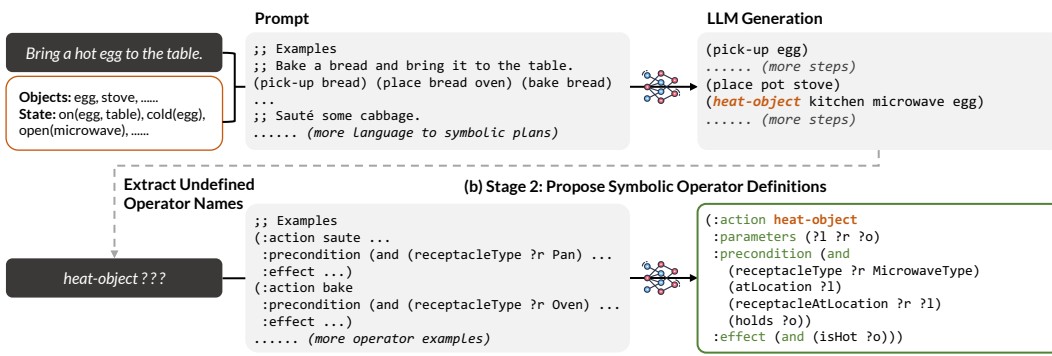

Figure 4: Our two-stage prompting method for generating candidate operator definitions. (a) Given a task instruction, we first prompt an LLM to generate a candidate symbolic task decomposition. (b) We then extract undefined operator names that appear in the sequences and prompt an LLM to generate symbolic definitions.

## 3.1 OPERATOR PROPOSAL: $\mathcal{A}_i \leftarrow \mathcal{A}_{i-1} \cup \text{ProposeOperatorDefinitions}(\mathcal{P}, \{\ell_t\})$

At each iteration $i$, we use a pretrained LLM to extend the previous operator library $\mathcal{A}_{i-1}$ with a large set of candidate operator definitions proposed by the LLM based on the task language descriptions and environment features $\mathcal{P}$. This yields an extended candidate library $\mathcal{A}_i'$ where each $a \in \mathcal{A}_i' = \langle name, args, pre, eff \rangle$ where $name$ is a human-readable action name and $args, pre, eff$ are a PDDL operator definition. We employ a two-stage prompting strategy: symbolic task decomposition followed by symbolic operator definition.

**Example.** Fig. 4 shows a concrete example. Given a task instruction (*Bring a hot egg to the table*) and the abstract state description, we first prompt the LLM to generate an abstract task decomposition, which may contain operator names that are undefined in the current operator library. Next, we extract the names of those undefined operators and prompt LLMs to generate the actual symbolic operator descriptions, in this case, the new *heat-object* operator.

**Symbolic task decomposition.** For a given task $\ell_t$ and a initial state $x_0$, we first translate the raw state $x_0$ into a symbolic description $\Phi(x_0)$. To constrain the length of the state description, we only include unary features in the abstract state (i.e., only object categories and properties). Subsequently, we present a few-shot prompt to the LLM and query it to generate a proposed task decomposition conditioned on the language description $\ell_t$. It generates a sequence of named high-level actions and their arguments, which explicitly can include high-level actions that are not yet defined in the current action library. We then extract all the operator names proposed across tasks as the candidate high-level operators. Note that while in principle we might use the LLM-proposed task decomposition itself as a high-level plan, we find empirically that this is less accurate and efficient than a formal planner.

**Symbolic operator definition.** With the proposed operator names and their usage examples (i.e., the actions and their arguments in the proposed plans), we then few-shot prompt the LLM to generate candidate operator *definitions* in the PDDL format (argument types, and pre/postconditions defined based on features in $\mathcal{P}$). We also post-process the generated operator definitions to remove feature

names not present in $\mathcal{P}$ and correct syntactic errors. We describe implementation details for our syntax correction strategy in the appendix.

## 3.2 Goal Proposal and Planning: $result^{(j)} \leftarrow \text{Execute}(x_0^{(j)}, \text{BiLevelPlan}(\mathcal{A}_i, \ell_t^{(j)}, \pi))$

At each iteration $i$, we then attempt to *BiLevelPlan* for unsolved tasks in the dataset. This step attempts to find and execute a low-level action sequence $\{u_1, u_2, \cdots, u_H\} \in \mathcal{U}^H$ for each task using the proposed operators in $\mathcal{A}_i'$ that satisfies the unknown goal predicate $g_t$ for each task. This provides the environment reward signal for action learning. Our *BiLevelPlan* has three steps.

**Symbolic goal proposal:** As defined in Sec. 2, each task is associated with a queryable but unknown goal predicate $g_t$ that can be represented as a first-order logic formula $f_t$ over symbolic features in $\mathcal{P}$. Our agent only has access to a linguistic task description $\ell_t$, so we use a few-shot prompted LLM to predict candidate goal formulas $F_t'$ conditioned on $\ell_t$ and features $\mathcal{P}$.

**High-level planning**: Given each candidate goal formula $f_t' \in F_t'$, the initial abstract problem state $s_0$, and the current candidate operator library $\mathcal{A}'$, we search for a *high-level plan* $P_A = \{(a_1, o_{1_i}...), \cdots, (a_K, o_{K_i}...)\}$ as a sequence of high-level actions from $\mathcal{A}'$ concretized with object arguments $o$, such that executing the action sequence would satisfy $f_t'$ according to the operator definitions. This is a standard symbolic PDDL planning formulation; we use an off-the-shelf symbolic planner, FastDownward (Helmert, 2006) to find high-level plans.

**Low-level planning and environment feedback**: We then search for a low-level plan as a sequence of low-level actions $\{u_1, u_2, \cdots, u_H\} \in \mathcal{U}^H$, conditioned on the high-level plan structure. Each concretized action tuple $(a_i, o_{1_i}...) \in P_A$ defines a local subgoal $sg_i$, as the operator postcondition parameterized by the object arguments $o$. For each $(a_i, o_{1_i}...) \in P_A$, we therefore search for a sequence of low-level actions $u_{i_1}, u_{i_2}...$ that satisfies the local subgoal $sg_i$. We search with a fixed budget per subgoal, and fail early if we are unable to satisfy the local subgoal $sg_i$. If we successfully find a complete sequence of low-level actions satisfying all local subgoals $sg_i$ in $P_A$, we execute all low-level actions and query the hidden goal predicate $g_t$ to determine environment reward. We implement a basic learning procedure to simultaneously learn subgoal-conditioned controllers over time (described in Section 3.3), but our formulation is general and supports many hierarchical planning schemes (such as sampling-based low-level planners (LaValle, 1998) or RL algorithms).

## 3.3 Low-Level Learning and Guided Search: $\theta \leftarrow \text{UpdateSubgoalPolicy}(\theta, result)$

The sequence of subgoals $sg_i$ corresponding to high-level plans $P_A$ already restricts the local low-level planning horizon. However, we further learn subgoal-conditioned low-level policies $\pi(u|x; sg)$ from environment feedback during training to accelerate low-level planning. To exploit shared structure across subgoals, we learn a shared controller for all operators from $x \in \mathcal{X}$ and conjunctions of predicates in $sg$. To maximize learning during training, we use a hindsight goal relabeling scheme (Andrychowicz et al., 2017), supervising on all conjunctions of predicates in the state as we roll out low-level search. While the shared controller could be learned as a supervised neural policy, we find that our learned operators sufficiently restrict the search to permit learning an even simpler count-based model from $X, sg \rightarrow u \in \mathcal{U}$. We provide additional details in the Appendix.

## 3.4 Scoring LLM Operator Proposals: $\mathcal{A}_i \leftarrow \text{ScoreAndFilter}(\mathcal{A}_i, result)$

Finally, we update the learned operator library $\mathcal{A}_i$ to retain candidate operators that were useful and successful in bi-level planning. Concretely, we estimate operator candidate $a_i' \in \mathcal{A}_i'$ accuracy across the bi-level plan executions as $s/b$ where $b$ counts the total times $a_i'$ appeared in a high-level plan and $s$ counts successful execution of the corresponding low-level action sequence to achieve the subgoal associated with $a_i'$. We retain operators if $b > \tau_b$ and $s/b > \tau_r$, where $\tau_b, \tau_r$ are hyperparameters. Note that this scoring procedure learns whether operators are accurate and support low-level planning independently of whether the LLM-predicted goals $f_t'$ matched the true unknown goal predicates $g_t$.

## 4 Experiments

**Domains.** We evaluate our approach on two-language specified planning-benchmarks: *Mini Minecraft* and *ALFRED* (Shridhar et al., 2020). *Mini Minecraft* (Fig. 5, *top*) is a procedurally-generated Minecraft-like benchmark (Chen et al., 2021; Luo et al., 2023) on a 2D grid world that requires

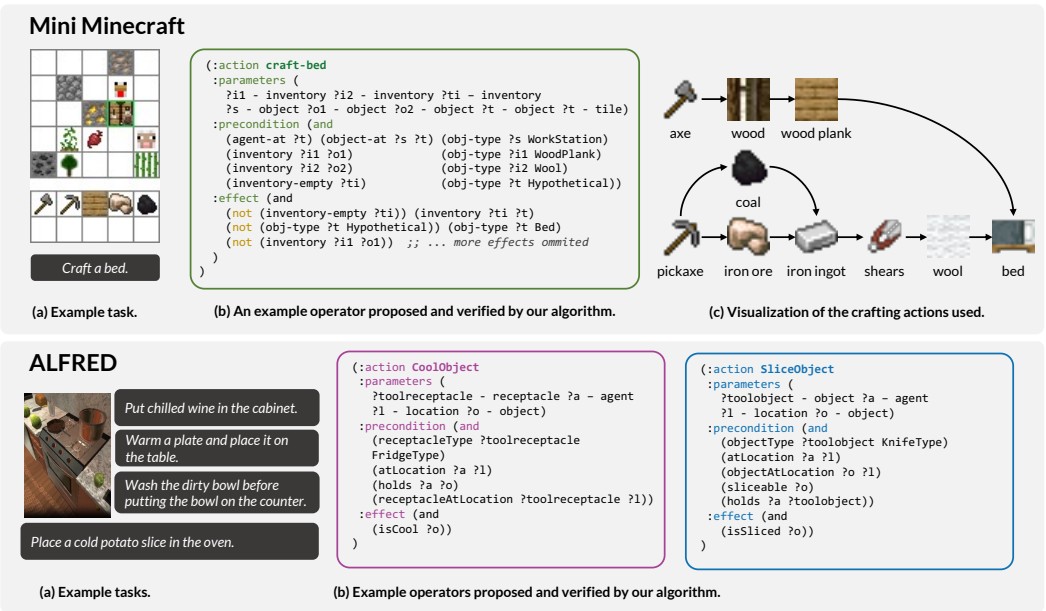

Figure 5: *Top*: (a) The Mini Minecraft environment, showing an intermediate step towards *crafting a bed*. (b) Operator proposed by an LLM and verified by our algorithm through planning and execution. (c) Low-level actions involved in crafting the bed. *Bottom*: (a) The ALFRED household environment. (b) Example operators proposed by LLM and verified by our algorithm, which are composed to solve the *cold potato slice* task.

complex, extended planning. The agent can use tools to mine resources and craft objects. The ability to create new objects that themselves permit new actions yields an enormous action space at each time step (>2000 actions) and very long-horizon tasks (26 high-level steps for the most complex task, without path-planning.) *ALFRED* (Fig. 5, *bottom*) is a household planning benchmark of human-annotated but formally verifiable tasks defined over a simulated Unity environment (Shridhar et al., 2020). The tasks include object rearrangements and those with object states such as heating and cleaning. Ground-truth high-level plans in the ALFRED benchmark compose 5-10 high-level operators, and low-level action trajectories have on average 50 low-level actions. There over 100 objects that the agent can interact with in each interactive environment. See the Appendix for details.

**Experimental setup.** We evaluate in an iterative continual learning setting; except on the compositional evaluations, we learn from *n=2* iterations through all (randomly ordered) tasks and report final accuracy on those tasks. All experiments and baselines use GPT-3.5. For each task, at each iteration, we sample *n=4* initial goal proposals and *n=4* initial task decompositions, and *n=3* operator definition proposals for each operator name. We report *best-of* accuracy, scoring a task as solved if verification passes on at least one of the proposed goals. For Minecraft, we set the motion planning budget for each subgoal to ≤1000 nodes. For ALFRED, which requires a slow Unity simulation, we set it to 50 nodes. Additional temperature and sampling details are in the Appendix.

We evaluate on three *Mini Minecraft* benchmark variations to test how our approach generalizes to complex, compositional goals. In the simplest **Mining** benchmark, all goals involve mining a target item from an appropriate initial resource with an appropriate tool (e.g., Mining *iron* from *iron_ore* with an *axe*). In the harder **Crafting** benchmark, goals involve crafting a target artifact (e.g., a *bed*), which may require mining a few target resources. The most challenging **Compositional** benchmark combines mining and crafting tasks, in environments that only begin with raw resources and two starting tools (axe and pickaxe). Agents may need to compose multiple skills to obtain other downstream resources (see Fig. 5 for an example). To test action generalization, we report evaluation on the *Compositional* using *only* actions learned previously in the **Mining** and **Crafting** benchmarks.

We similarly evaluate on an *ALFRED* benchmark of **Simple and Compositional** tasks drawn from the original task distribution in Shridhar et al. (2020). This distribution contains simple tasks that require picking up an object and placing it in a new location, picking up objects, applying a single household skill to an object and moving them to a new location (e.g., *Put a clean apple on the dining table*), and compositional tasks that require multiple skills (e.g., *Place a hot sliced potato on the*

| Mini Minecraft (n=3) | LLM Predicts? | Library? | Mining | Crafting | Compositional |
|---|---|---|---|---|---|
| Low-level Planning Only | Goal | ✗ | 31% ($\sigma$=0.0%) | 9% ($\sigma$=0.0%) | 9% ($\sigma$=0.0%) |
| Subgoal Prediction | Sub-goals | ✗ | 33% ($\sigma$=1.6%) | 36% ($\sigma$=5.6%) | 6% ($\sigma$=1.7%) |
| Code Policy Prediction | Sub-policies | ✓ | 15% ($\sigma$=1.2%) | 39% ($\sigma$=3.2%) | 10% ($\sigma$=1.7%) |
| Ada (Ours) | Goal+Operators | ✓ | 100% ($\sigma$=0.0%) | 100% ($\sigma$=7.5%) | 100% ($\sigma$=4.1%) |

| ALFRED (n=3 replications) | LLM Predicts? | Library? | Original (Simple + Compositional Tasks) | | |
|---|---|---|---|---|---|
| Low-level Planning Only | Goal | ✗ | 21% ($\sigma$=1.0%) | | |
| Subgoal Prediction | Sub-goal | ✗ | 2% ($\sigma$=0.4%) | | |
| Code Policy Prediction | Sub-policies | ✓ | 2% ($\sigma$=0.9%) | | |
| Ada (Ours) | Goal+Operators | ✓ | 79% ($\sigma$=0.9%) | | |

Table 1: (*Top*) Results on *Mini Minecraft*. Our algorithm successfully recovers all intermediate operators for mining and crafting, which enable generalization to more compositional tasks (which use up to 26 operators) without any additional learning. (*Bottom*) Results on ALFRED. Our algorithm recovers all required household operators, which generalize to more complex compositional tasks. All results report mean performance and STD from *n=3* random replications for all models.

*counter*). We use a random subset of n=223 tasks, selected from an initial 250 that we manually filter to remove completely misspecified goals (which omit any mention of the target object or skill).

**Baselines.** We compare our method to three baselines of language-guided planning.

***Low-level Planning Only*** uses an LLM to predict only the symbolic goal specification conditioned on the high-level predicates and linguistic goal, then uses the low-level planner to search directly for actions that satisfy that goal. This baseline implements a model like **LLM+P** (Liu et al., 2023), which uses LLMs to translate linguistic goals into planning-compatible formal specifications, then attempt to plan directly towards these with no additional representation learning.

***Subgoal Prediction*** uses an LLM to predict a sequence of high-level subgoals (as PDDL pre/postconditions with object arguments), conditioned on the high-level predicates, and task goal and initial environment state. This baseline implements a model like **SayCan** (Ahn et al., 2022), which uses LLMs to directly predict goal *and* a sequence of decomposed formal subgoal representations, then applies low-level planning over these formal subgoals.

***Code Policy Prediction*** uses an LLM to predict the definitions of a library of *imperative local code policies* in Python (with cases and control flow) over an imperative API that can query state and execute low-level actions.) Then, as FastDownward planning is no longer applicable, we also use the LLM to predict the function call sequences with arguments for each task. This baseline implements a model like **Voyager** (Wang et al., 2023a), which uses an LLM to predict a library of skills implemented as imperative code for solving individual tasks. Like Voyager, we verify the individual code skills during interactive planning, but do not use a more global learning objective to attempt to learn a concise or non-redundant library.

### 4.1 RESULTS

**What action libraries do we learn?** Fig. 5 shows example operators learned on each domain (Appendix A.3 contains the full libraries of operators learned on both domains from a randomly sampled run of the n=3 replications). In *Mini Minecraft*, we manually inspect the library and find that we learn operators that correctly specify the appropriate tools, resources, and outputs for all intermediate mining actions (on **Mining**) and crafting actions (on **Crafting**), allowing perfect direct generalization to the **Compositional** tasks without any additional training on these complex tasks. In *ALFRED*, we compare the learned libraries from all runs to the ground-truth operator library hand-engineered in Shridhar et al. (2020). The ground-truth operator set contains 8 distinct operators corresponding to different compositional skills (e.g., *Slicing*, *Heating*, *Cleaning*, *Cooling*). Across all replications, model reliably recovers semantically identical (same predicate preconditions and postconditions) definitions for *all* of these ground-truth operators, except for a single operator that is defined disjunctively (the ground-truth *Slice* skill specifies either of two types of knives), which we occasionally learn as two distinct operators or only recover with one of these two types.

We also inspect the learning trajectory and find that, through the interactive learning loop, we successfully *reject* many initially proposed operator definitions sampled from the language model that turn out to be redundant (which would make high-level planning inefficient), inaccurate (including apriori reasonable proposals that do not fit the environment specifications, such as proposing to *clean* objects with just a *towel*, when our goal verifiers require washing them with water in a *sink*), or

underspecified (such as those that omit key preconditions, yielding under-decomposed high-level task plans that make low-level planning difficult).

**Do these actions support complex planning and generalization?** Table 2 shows quantitative results from *n=3* randomly-initialized replications of all models, to account for random noise in sampling from the language model and stochasticity in the underlying environment (ALFRED). On Minecraft, where goal specification is completely clear due to the synthetic language, we solve all tasks in each evaluation variation, including the challenging *Compositional* setting — the action libraries learned from simpler mining/crafting tasks generalize completely to complex tasks that require crafting all intermediate resources and tools from scratch. On ALFRED, we vastly outperform all other baselines, demonstrating that the learned operators are much more effective for planning and compose generalizably to more complex tasks. We qualitatively find that failures on ALFRED occur for several reasons. One is *goal misspecification*, when the LLM does not successfully recover the formal goal predicate (often due to ambiguity in human language), though we find that on average, 92% of the time, the ground truth goal appears as one of the top-4 goals translated by the LLM. We also find failures due to low-level *policy inaccuracy*, when the learned policies fail to account for low-level, often geometric details of the environment (e.g., the learned policies are not sufficiently precise to place a tall bottle on an appropriately tall shelf). More rarely, we see planning failures caused by slight *operator overspecification* (e.g., the *Slice* case discussed above, in which we do not recover the specific disjunction over possible knives that can be used to slice.) Both operator and goal specification errors could be addressed in principal by sampling more (and more diverse) proposals.

**How does our approach compare to using the LLM to predict just goals, or predict task sequences?** As shown in Table 2, our approach vastly outperforms the **Low-level Planning Only** baseline on both domains, demonstrating the value of the action library for longer horizon planning. We also find a substantial improvement over the **Subgoal Prediction** baseline. While the LLM frequently predicts important high-level aspects of the task subgoal structure (as it does to propose operator definitions), it frequently struggles to robustly sequence these subgoals and predict appropriate concrete object groundings that correctly obey the initial problem conditions or changing environment state. These errors accumulate over the planning horizon, reflected in decreasing accuracy on the compositional Minecraft tasks (on ALFRED, this baseline struggles to solve any more than the basic pick-and-place tasks, as the LLM struggles to predict subgoals that accurately track whether objects are in appliances or whether the agent's single gripper is full with an existing tool.)

**How does our approach compare to using the LLM to learn and predict plans using imperative code libraries?** Somewhat surprisingly, we find that the *Code Policy* prediction baseline performs unevenly and often very poorly on our benchmarks. (We include additional results in A.2.1 showing that our model also dramatically outperforms this baseline using GPT-4 as the base LLM.) We find several key reasons for the poor performance of this baseline relative to our model, each which validate the key conceptual contributions of our approach. First, the baseline relies on the LLM as the planner – as the skills are written as general Python functions, rather than any planner-specific representation, we do not use an optimized planner like FastDownward. As with *Subgoal Prediction*, we find that the LLM is not a consistent or accurate planner. While it retrieves generally relevant skills from the library for each task, it often struggles to sequence them accurately or predict appropriate arguments given the initial problem state. Second, we find that imperative code is less suited in general as a hierarchical planning representation for these domains than the high-level PDDL and low-level local policy search representation we use in our model. This is because it uses control flow to account for environment details that would otherwise be handled by local search relative to a high-level PDDL action. Finally, our model specifically frames the library learning objective around learning a compact library of skills that enables efficient planning, whereas our Voyager re-implementation (as in Wang et al. (2023a)) simply grows a library of skills which are individually executable and can be used to solve individual, shorter tasks. Empirically, as with the original model in Wang et al. (2023a), this baseline learns *hundreds* of distinct code definitions on these datasets, which makes it harder to accurately plan and generalize to more complex tasks. Taken together, these challenges support our overarching library learning objective for hierarchical planning.

## 5 RELATED WORK

**Planning for language goals.** A large body of recent work attempts to use LLMs to solve planning tasks specified in language. One approach is to directly predict action sequences (Huang et al., 2022; Valmeekam et al., 2022; Silver et al., 2022; Wang et al., 2023b), but this has yielded mixed

results as LLMs can struggle to generalize or produce correct plans as problems grow more complex. To combat this, one line of work has explored structured and iterative prompting regimes (e.g., 'chain-of-thought' and feedback) (Mu et al., 2023; Silver et al., 2023; Zhu et al., 2023). Increasingly, other neuro-symbolic work uses LLMs to predict formal goal or action representations that can be verified or solved with symbolic planners (Song et al., 2023; Ahn et al., 2022; Xie et al., 2023; Arora & Kambhampati, 2023). These approaches leverage the benefits of a known planning domain model. Our goal in this paper is to leverage language models to *learn* this domain model. Another line of research aims at using LLMs to generate formal planning domain models for specific problems (Liu et al., 2023) and subsequently uses classical planners to solve the task. However, they are not considering generating grounded or hierarchical actions in an environment and not learning a library of operators that can be reused across different tasks. More broadly, we share the broad goal of building agents that can understand language and execute actions to achieve goals (Tellex et al., 2011; Misra et al., 2017; Nair et al., 2022). See also Luketina et al. (2019) and Tellex et al. (2020).

**Learning planning domain and action representations from language.** Another group of work has been focusing on learning latent action representations from language (Corona et al., 2021; Andreas et al., 2017; Jiang et al., 2019; Sharma et al., 2022; Luo et al., 2023). Our work differs from them in that we are learning a planning-compatible action abstraction from LLMs, instead of relying on human demonstrations and annotated step-by-step instructions. The more recent Wang et al. (2023a) adopts a similar overall problem specification, to learn libraries of actions as imperative code-based policies. Our results show that learning planning abstractions enables better integration with hierarchical planning, and, as a result, better performance and generalization to more complex problems. Other recent work (Nottingham et al., 2023) learns an environment model from interactive experience, represented as a task dependency graph; we seek to learn a richer state transition model (which represents the effects of actions) decomposed as operators that can be formally composed to verifiably satisfy arbitrarily complex new goals. Guan et al. (2024), published concurrently, seeks to learn PDDL representations; we show how these can be grounded hierarchically.

**Language and code.** In addition to Wang et al. (2023a), a growing body of work in program synthesis, both by learning lifted program abstractions that compress longer existing or synthesized programs (Bowers et al., 2023; Ellis et al., 2023; Wong et al., 2021; Cao et al., 2023). These approaches (including Wang et al. (2023a)) generally learn libraries defined over imperative and functional programming languages, such as LISP and Python. Our work is closely inspired by these and seeks to learn representations suited specifically to solving long-range planning problems.

**Hierarchical planning abstractions.** The hierarchical planning knowledge that we learn from LLMs and interactions in the environments are related to hierarchical task networks (Erol et al., 1994; Nejati et al., 2006), hierarchical goal networks (Alford et al., 2016), abstract PDDL domains (Konidaris et al., 2018; Bonet & Geffner, 2020; Chitnis et al., 2021; Asai & Muise, 2020; Mao et al., 2022; 2023), and domain control knowledge (de la Rosa & McIlraith, 2011). Most of these approaches require manually specified hierarchical planning abstractions; others learn them from demonstrations or interactions. By contrast, we leverage human language to guide the learning of such abstractions.

## 6 DISCUSSION AND FUTURE WORK

Our evaluations suggest a powerful role for language within AI systems that form complex, long-horizon plans — as a rich source of background knowledge about the right *action abstractions* for everyday planning domains, which contains broad human priors about environments, task decompositions, and potential future goals. A core goal of this paper was to demonstrate how to integrate this knowledge into the search, grounding, and verification toolkits developed in hierarchical planning.

We leave open many possible extensions towards future work. Key **limitations** of our current framework point towards important directions for further integrating LMs and hierarchical planning to scale our approach: here, we build on an existing set of pre-defined symbolic predicates for initially representing the environment state; do not yet tackle fine-grained, geometric motor planning; and use a general LLM (rather than one fine-tuned for extended planning). **Future work** might generally tackle these problems by further asking how else linguistic knowledge and increasingly powerful or multimodal LLMs could be integrated here: to *propose* useful named predicates over initial perceptual inputs (e.g., images) (Migimatsu & Bohg, 2022); or to speed planning by bootstrapping hierarchical planning abstractions using the approach here, but then to progressively transfer planning to another model, including an LLM, to later compose and use the learned representations.

**Acknowledgement.** We thank anonymous reviewers for their valuable comments. We gratefully acknowledge support from ONR MURI grant N00014-16-1-2007; from the Center for Brain, Minds, and Machines (CBMM, funded by NSF STC award CCF-1231216); from NSF grant 2214177; from NSF grant CCF-2217064 and IIS-2212310; from Air Force Office of Scientific Research (AFOSR) grant FA9550-22-1-0249; from ONR MURI grant N00014-22-1-2740; from ARO grant W911NF-23-1-0034; from the MIT-IBM Watson AI Lab; from the MIT Quest for Intelligence; from Intel; and from the Boston Dynamics Artificial Intelligence Institute. Any opinions, findings, and conclusions or recommendations expressed in this material are those of the authors and do not necessarily reflect the views of our sponsors.

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

# A APPENDIX

We will release a complete code repository containing our full algorithm implementation, all baselines, and benchmark tasks. Here, we provide additional details on our implementational choices.

## A.1 BENCHMARKS

*Mini Minecraft* (Fig. 5, *top*) is a procedurally-generated Minecraft-like benchmark (Chen et al., 2021; Luo et al., 2023) that requires complex, extended planning. The environment places an agent on a 2D map containing various resources, tools, and crafting stations. The agent can use appropriate tools to mine new items from raw resources (e.g. use an *axe* to obtain *wood* from *trees*), or collect resources into an inventory to craft new objects (e.g. combining *sticks* and *iron ingots* to craft a *sword*, which itself can be used to obtain *feathers* from a *chicken*). The ability to create new objects that themselves permit new actions yields an enormous action space at each time step (>2000 actions, considering different combinations of items to use) and very long-horizon tasks (26 steps for the most complex task, even without path-planning.) The provided environment predicates allow querying object types and inventory contents. Low-level actions allow the agent to move and apply tools to specific resources. To focus on complex crafting, we provide a low-level *move-to* action to move directly to specified locations. Linguistic goal specifications are synthetically generated from a simple grammar over craftable objects and resources (e.g. *Craft a sword*, *Mine iron ore*).

*ALFRED* (Fig. 5, *bottom*) is a household planning benchmark of human-annotated but formally verifiable tasks defined over a simulated Unity environment (Shridhar et al., 2020). The interactive environment places an agent in varying 3D layouts, each containing appliances and dozens of household objects. The provided environment includes predicates for querying object types, object and agent locations, and classifiers over object states (eg. whether an object is *hot* or *on*). Low-level actions enable the agent to pick up and place objects, apply tools to other objects, and open, close, and turn on appliances. As specified in Shridhar et al. (2020), ground-truth high-level plans in the ALFRED benchmark compose 5-10 high-level operators, and low-level action trajectories have on average 50 low-level actions. There over 100 objects that the agent can interact with in each interactive environment.

As with Minecraft, we provide a low-level method to move the agent directly to specified locations. While ALFRED is typically used to evaluate detailed instruction following, we focus on a *goal-only* setting that only uses the goal specifications. The human-annotated goals introduce ambiguity, underspecification, and errors with respect to the ground-truth verifiable tasks (eg. people refer to *tables* without specifying if they mean the *side table*, *dining table*, or *desk*; a *light* when there are multiple distinct lamps; or a *cabbage* when they want *lettuce*).

## A.2 ADDITIONAL METHODS IMPLEMENTATION DETAILS

### A.2.1 LLM PROMPTING

We use `gpt-3.5-turbo-16k` for all experiments and baselines. Here, we describe the contents of the LLM few-shot prompts used in our method in more detail. **Symbolic Task Decomposition** For all unsolved tasks, at each iteration, we sample a set of symbolic task descriptions as a sequence of named high-level actions and their arguments. We construct a few-shot prompt consisting of the following components:

1. A brief natural language header (*;;;; Given natural language goals, predict a sequence of PDDL actions*);

2. A sequence of example $(l_t, P_A)$ tuples containing linguistic goals and example task decompositions. To avoid biasing the language model in advance, we provide example task decompositions for similar, constructed tasks that do not use any of the skills that need to be learned in our two domains.

   For example, on ALFRED, these example task decompositions are for example tasks (*bake a potato and put it in the fridge*, *place a baked, grated apple on top of the dining table*, *place a plate in a full sink.*, and *pick up a laptop and then carry it over to the desk lamp, then restart the desk lamp.*), and our example task decompositions suggest named operators

*BakeObject*, *GrateObject*, *FillObject*, and *RestartObject*, none of which appear in the actual training set.

3. At iterations $> 0$, we also provide a sequence of sampled $(l_t, P_A)$ tuples randomly sampled from any solved tasks and their discovered high-level plans. This means that few-shot prompting better represents the true task distribution over successive iterations.

In our experiments, we prompt with temperature=1.0 and draw n=4 task decomposition samples per unsolved task.

**Symbolic Operator Definition** For all unsolved tasks, at each iteration, we sample proposed operator definitions consisting of *args, pre, eff* conditioned on all undefined operator names that appear in the proposed task decompositions.

For each operator name, we construct a few-shot prompt consisting of the following components:

1. A brief natural language header (*You are a software engineer who will be writing planning operators in the PDDL planning language. These operators are based on the following PDDL domain definition.*

2. The full set of environment predicates vocabulary of high-level environment predicates $\mathcal{P}$, as well as valid named argument values (eg. object types).

3. A sequence of example *name, args, pre, eff* operator definitions demonstrating the PDDL definition format. As with task decomposition, of course, we do not provide any example operator definitions that we wish to learn from our dataset.

4. At iterations $> 0$, we include as many possible validated *name, args, pre, eff* operators defined in the current library (including new learned operators). If there are shared patterns between operators, this means that few-shot prompting also better represents the true operator structure over successive iterations.

In our experiments, we prompt with temperature=1.0 and draw n=3 task decomposition samples per unsolved task. However, in our pilot experiments, we actually find that sampling directly from the token probabilities defined by this few-shot prompt does not produce sufficiently diverse definitions for each operator name. We instead directly prompt the LLM to produce up to N distinct operator definitions sequentially.

We find that GPT 3.5 frequently produces syntactically invalid operator proposals – proposed operators often include invent predicates and object types that are not defined in the environment vocabulary, do not obey the predicate typing rules, or do not have the correct number and types of arguments. While this might improve with finetuned or larger LLMs, we instead implement a simple post-processing heuristic to correct operators with syntactic errors, or reject operators altogether: as operator pre and postconditions are represented as conjunctions of predicates, we remove any invalid predicates (predicates that are invented or that specify invalid arguments); we collect all arguments named across the predicates and use the ground truth typing to produce the final *args*, and we reject any operators that have 0 valid postcondition predicates. This post-processing procedure frequently leaves operators underspecified (e.g., the resulting operators now are missing necessary preconditions, which were partially generated but syntactically incorrect in the proposal); we allow our full operator learning algorithm to verify and reject these operators.

**Symbolic Goal Proposal** Finally, as described in 3.2, we also use an LLM to propose a set of candidate goal definitions as FOL formulas $F'_t$ defined over the environment predicates $\mathcal{P}$ for each task. Our prompting technique is very similar to that used in the rest of our algorithm. For each task, we we construct a few-shot prompt consisting of the following components:

1. A brief natural language header (*You are a software engineer who will be writing goal definitions for a robot in the PDDL planning language.*

2. The full set of environment predicates vocabulary of high-level environment predicates $\mathcal{P}$, as well as valid named argument values (eg. object types).

3. A sequence of example $l_t, f_t$ language and FOL goal formulas. In our experiments, during training, unlike in the previous prompts (where including ground truth operators would solve the learning problem), we *do* sample an initial set of goal definitions from the training

| *Mini Minecraft* | LLM Predicts? | Library? | Mining | Crafting | Compositional |
|---|---|---|---|---|---|
| Code Policy Prediction | Sub-policies | ✓ | 12% | 37% | 11% |
| Ours | Goal+Operators | ✓ | 100% | 100% | 100% |

| *ALFRED* (*n=3 replications*) | LLM Predicts? | Library? | Original (Simple + Compositional Tasks) | | |
|---|---|---|---|---|---|
| Code Policy Prediction | Sub-policies | ✓ | 11% | | |
| Ours | Goal+Operators | ✓ | 70% | | |

Table 2: **Results with GPT-4 as the LLM backbone**: On both *Mini Minecraft* (*Top*) and ALFRED (*Bottom*), our algorithm recovers all required operators, which generalize to more complex compositional tasks. Switching to GPT-4 does not impact performance trends observed across the *Code as Policies (Voyager)* baseline and our method.

distribution as our initial example supervision. We set supervision to a randomly sampled fraction (0.1) of the training distribution.

4. At iterations $> 0$, we also include $l_t$, $f_t$ examples from successfully solved tasks.

In our experiments, we prompt with temperature=1.0 and draw n=4 task decomposition samples per unsolved task. As with the operator proposal, we also find that sampling directly from the token probabilities defined by this few-shot prompt does not produce sufficiently diverse definitions for each linguistic goal to correct for ambiguity in the human language (eg. to define the multiple concrete *Table* types that a person might mean when referring to a *table*). We therefore again instead directly prompt the LLM to produce up to N distinct operator definitions sequentially.

We also post-process proposed goals using the same syntactic criterion to remove invalid predicates in the FOL formula, and reject any empty goals.

### A.2.2 POLICY LEARNING AND GUIDED LOW-LEVEL SEARCH

Concretely, we implement our policy-guided low-level action search as the following. We maintain a dictionary $D$ that maps subgoals (a conjunction of atoms) to a set of candidate low-level action trajectories. When planning for a new subgoal $sg$, if $D$ contains the trajectory, we prioritize trying candidate low-level trajectories in $D$. Otherwise, we fall back to a brute-force breadth-first search over all possible action trajectories. To populate $D$, during the BFS, we compute the difference in the environment state before and after the agent executes any sampled trajectory and the corresponding trajectory $t$ that caused the state change. Here the state difference can be viewed as a subgoal $sg$ achieved by executing $t$. Rather than directly adding the $(sg, t)$ as a key-value pair to $D$, we *lift* the trajectory and environment state change by replacing concrete objects in $sg$ and $t$ by variables. Note that we update $D$ with each sampled trajectory in the BFS even if it doesn't achieve the subgoal specified in the BFS search.

When the low-level search receives a subgoal $sg$, we again lift it by replacing objects with variables, and try to match it with entries in $D$. If $D$ contains multiple trajectories $t$ for a given subgoal $sg$, we track how often a given trajectory succeeds for a subgoal and prioritize trajectories with the most successes.

### A.3 EXPERIMENTS

**Learned Operator Libraries on Minecraft** The following shows the full PDDL domain definition including the initial provided vocabulary of symbolic environment constants and predicates, initial pick and place operators and example operator, and all ensuing learned operators combined from the **Mining** and **Crafting** benchmarks.

```
1  (define (domain crafting-world-v20230404-teleport)
2  (:requirements :strips)
3  (:types
4      tile
5      object
6      inventory
7      object-type
8  )
9  (:constants
```

```
10      Key − object−type
11      WorkStation − object−type
12      Pickaxe − object−type
13      IronOreVein − object−type
14      IronOre − object−type
15      IronIngot − object−type
16      CoalOreVein − object−type
17      Coal − object−type
18      GoldOreVein − object−type
19      GoldOre − object−type
20      GoldIngot − object−type
21      CobblestoneStash − object−type
22      Cobblestone − object−type
23      Axe − object−type
24      Tree − object−type
25      Wood − object−type
26      WoodPlank − object−type
27      Stick − object−type
28      Sword − object−type
29      Chicken − object−type
30      Feather − object−type
31      Arrow − object−type
32      Shears − object−type
33      Sheep − object−type
34      Wool − object−type
35      Bed − object−type
36      Boat − object−type
37      SugarCanePlant − object−type
38      SugarCane − object−type
39      Paper − object−type
40      Bowl − object−type
41      PotatoPlant − object−type
42      Potato − object−type
43      CookedPotato − object−type
44      BeetrootCrop − object−type
45      Beetroot − object−type
46      BeetrootSoup − object−type
47
48      Hypothetical − object−type
49      Trash − object−type
50   )
51   (:predicates
52      (tile−up ?t1 − tile ?t2 − tile)
53      (tile−down ?t1 − tile ?t2 − tile)
54      (tile−left ?t1 − tile ?t2 − tile)
55      (tile−right ?t1 − tile ?t2 − tile)
56
57      (agent−at ?t − tile)
58      (object−at ?x − object ?t − tile)
59      (inventory−holding ?i − inventory ?x − object)
60      (inventory−empty ?i − inventory)
61
62      (object−of−type ?x − object ?ot − object−type)
63   )
64
65   (:action move−to
66      :parameters (?t1 − tile ?t2 − tile)
67      :precondition (and (agent−at ?t1))
68      :effect (and (agent−at ?t2) (not (agent−at ?t1)))
69   )
70   (:action pick−up
71      :parameters (?i − inventory ?x − object ?t − tile)
72      :precondition (and (agent−at ?t) (object−at ?x ?t) (inventory−empty ?i)
          )
```

```
73    : effect (and (inventory-holding ?i ?x) (not (object-at ?x ?t)) (not (
         inventory-empty ?i)))
74    )
75    (:action place-down
76     :parameters (?i - inventory ?x - object ?t - tile)
77     :precondition (and (agent-at ?t) (inventory-holding ?i ?x))
78     :effect (and (object-at ?x ?t) (not (inventory-holding ?i ?x)) (
         inventory-empty ?i))
79    )
80    (:action mine-iron-ore
81     :parameters (?toolinv - inventory ?targetinv - inventory ?x - object ?
         tool - object ?target - object ?t - tile)
82     :precondition (and
83       (agent-at ?t)
84       (object-at ?x ?t)
85       (object-of-type ?x IronOreVein)
86       (inventory-holding ?toolinv ?tool)
87       (object-of-type ?tool Pickaxe)
88       (inventory-empty ?targetinv)
89       (object-of-type ?target Hypothetical)
90     )
91     :effect (and
92       (not (inventory-empty ?targetinv))
93       (inventory-holding ?targetinv ?target)
94       (not (object-of-type ?target Hypothetical))
95       (object-of-type ?target IronOre)
96     )
97    )
98    (:action mine-wood_2
99     :parameters (?t - tile ?x - object ?toolinv - inventory ?tool - object ?
         targetinv - inventory ?target - object)
100
101    :precondition (and
102      (agent-at ?t)
103      (object-at ?x ?t)
104      (object-of-type ?x Tree)
105      (inventory-holding ?toolinv ?tool)
106      (object-of-type ?tool Axe)
107      (inventory-empty ?targetinv)
108      (object-of-type ?target Hypothetical)
109    )
110    :effect (and
111      (not (inventory-empty ?targetinv))
112      (inventory-holding ?targetinv ?target)
113      (not (object-of-type ?target Hypothetical))
114      (object-of-type ?target Wood)
115    )
116   )
117   (:action mine-wool1_0
118    :parameters (?t - tile ?x - object ?toolinv - inventory ?tool - object ?
         targetinv - inventory ?target - object)
119
120    :precondition (and
121      (agent-at ?t)
122      (object-at ?x ?t)
123      (object-of-type ?x Sheep)
124      (inventory-holding ?toolinv ?tool)
125      (object-of-type ?tool Shears)
126      (inventory-empty ?targetinv)
127      (object-of-type ?target Hypothetical)
128    )
129    :effect (and
130      (not (inventory-empty ?targetinv))
131      (inventory-holding ?targetinv ?target)
132      (not (object-of-type ?target Hypothetical))
```

```
133       (object-of-type ?target Wool)
134     )
135   )
136   (:action mine-potato_0
137     :parameters (?t - tile ?x - object ?targetinv - inventory ?target -
          object)
138
139     :precondition (and
140       (agent-at ?t)
141       (object-at ?x ?t)
142       (object-of-type ?x PotatoPlant)
143       (inventory-empty ?targetinv)
144       (object-of-type ?target Hypothetical)
145     )
146     :effect (and
147       (not (inventory-empty ?targetinv))
148       (inventory-holding ?targetinv ?target)
149       (not (object-of-type ?target Hypothetical))
150       (object-of-type ?target Potato)
151     )
152   )
153   (:action mine-sugar-cane_2
154     :parameters (?t - tile ?x - object ?toolinv - inventory ?tool - object ?
          targetinv - inventory ?target - object)
155
156     :precondition (and
157       (agent-at ?t)
158       (object-at ?x ?t)
159       (object-of-type ?x SugarCanePlant)
160       (inventory-holding ?toolinv ?tool)
161       (object-of-type ?tool Axe)
162       (inventory-empty ?targetinv)
163       (object-of-type ?target Hypothetical)
164     )
165     :effect (and
166       (not (inventory-empty ?targetinv))
167       (inventory-holding ?targetinv ?target)
168       (not (object-of-type ?target Hypothetical))
169       (object-of-type ?target SugarCane)
170     )
171   )
172   (:action mine-beetroot_1
173     :parameters (?t - tile ?x - object ?toolinv - inventory ?tool - object ?
          targetinv - inventory ?target - object)
174
175     :precondition (and
176       (agent-at ?t)
177       (object-at ?x ?t)
178       (object-of-type ?x BeetrootCrop)
179       (inventory-holding ?toolinv ?tool)
180       (inventory-empty ?targetinv)
181       (object-of-type ?target Hypothetical)
182     )
183     :effect (and
184       (not (inventory-empty ?targetinv))
185       (inventory-holding ?targetinv ?target)
186       (not (object-of-type ?target Hypothetical))
187       (object-of-type ?target Beetroot)
188     )
189   )
190   (:action mine-feather_1
191     :parameters (?t - tile ?x - object ?toolinv - inventory ?tool - object ?
          targetinv - inventory ?target - object)
192
193     :precondition (and
```

```
194        (agent-at ?t)
195        (object-at ?x ?t)
196        (object-of-type ?x Chicken)
197        (inventory-holding ?toolinv ?tool)
198        (object-of-type ?tool Sword)
199        (inventory-empty ?targetinv)
200        (object-of-type ?target Hypothetical)
201      )
202      :effect (and
203        (not (inventory-empty ?targetinv))
204        (inventory-holding ?targetinv ?target)
205        (not (object-of-type ?target Hypothetical))
206        (object-of-type ?target Feather)
207      )
208    )
209    (:action mine-cobblestone_2
210      :parameters (?t - tile ?x - object ?toolinv - inventory ?tool - object ?
           targetinv - inventory ?target - object)
211
212      :precondition (and
213        (agent-at ?t)
214        (object-at ?x ?t)
215        (object-of-type ?x CobblestoneStash)
216        (inventory-holding ?toolinv ?tool)
217        (object-of-type ?tool Pickaxe)
218        (inventory-empty ?targetinv)
219        (object-of-type ?target Hypothetical)
220      )
221      :effect (and
222        (not (inventory-empty ?targetinv))
223        (inventory-holding ?targetinv ?target)
224        (not (object-of-type ?target Hypothetical))
225        (object-of-type ?target Cobblestone)
226      )
227    )
228    (:action mine-gold-ore1_2
229      :parameters (?t - tile ?x - object ?toolinv - inventory ?tool - object ?
           targetinv - inventory ?target - object)
230
231      :precondition (and
232        (agent-at ?t)
233        (object-at ?x ?t)
234        (object-of-type ?x GoldOreVein)
235        (inventory-holding ?toolinv ?tool)
236        (object-of-type ?tool Pickaxe)
237        (inventory-empty ?targetinv)
238        (object-of-type ?target Hypothetical)
239      )
240      :effect (and
241        (not (inventory-empty ?targetinv))
242        (inventory-holding ?targetinv ?target)
243        (not (object-of-type ?target Hypothetical))
244        (object-of-type ?target GoldOre)
245      )
246    )
247    (:action mine-coal1_0
248      :parameters (?t - tile ?x - object ?toolinv - inventory ?tool - object ?
           targetinv - inventory ?target - object)
249
250      :precondition (and
251        (agent-at ?t)
252        (object-at ?x ?t)
253        (object-of-type ?x CoalOreVein)
254        (inventory-holding ?toolinv ?tool)
255        (object-of-type ?tool Pickaxe)
```

```
256        (inventory-empty ?targetinv)
257        (object-of-type ?target Hypothetical)
258    )
259    :effect (and
260      (not (inventory-empty ?targetinv))
261      (inventory-holding ?targetinv ?target)
262      (not (object-of-type ?target Hypothetical))
263      (object-of-type ?target Coal)
264    )
265  )
266  (:action mine-beetroot1_0
267    :parameters (?t - tile ?x - object ?targetinv - inventory ?target -
         object)
268
269    :precondition (and
270      (agent-at ?t)
271      (object-at ?x ?t)
272      (object-of-type ?x BeetrootCrop)
273      (inventory-empty ?targetinv)
274      (object-of-type ?target Hypothetical)
275    )
276    :effect (and
277      (not (inventory-empty ?targetinv))
278      (inventory-holding ?targetinv ?target)
279      (not (object-of-type ?target Hypothetical))
280      (object-of-type ?target Beetroot)
281    )
282  )
283  (:action craft-wood-plank
284    :parameters (?ingredientinv1 - inventory ?targetinv - inventory ?
         station - object ?ingredient1 - object ?target - object ?t - tile)
285    :precondition (and
286      (agent-at ?t)
287      (object-at ?station ?t)
288      (object-of-type ?station WorkStation)
289      (inventory-holding ?ingredientinv1 ?ingredient1)
290      (object-of-type ?ingredient1 Wood)
291      (inventory-empty ?targetinv)
292      (object-of-type ?target Hypothetical)
293    )
294    :effect (and
295      (not (inventory-empty ?targetinv))
296      (inventory-holding ?targetinv ?target)
297      (not (object-of-type ?target Hypothetical))
298      (object-of-type ?target WoodPlank)
299      (not (inventory-holding ?ingredientinv1 ?ingredient1))
300      (inventory-empty ?ingredientinv1)
301      (not (object-of-type ?ingredient1 Wood))
302      (object-of-type ?ingredient1 Hypothetical)
303    )
304    )
305  (:action craft-arrow
306    :parameters (?ingredientinv1 - inventory ?ingredientinv2 - inventory ?
         targetinv - inventory ?station - object ?ingredient1 - object ?
         ingredient2 - object ?target - object ?t - tile)
307    :precondition (and
308      (agent-at ?t)
309      (object-at ?station ?t)
310      (object-of-type ?station WorkStation)
311      (inventory-holding ?ingredientinv1 ?ingredient1)
312      (object-of-type ?ingredient1 Stick)
313      (inventory-holding ?ingredientinv2 ?ingredient2)
314      (object-of-type ?ingredient2 Feather)
315      (inventory-empty ?targetinv)
316      (object-of-type ?target Hypothetical)
```

```
317      )
318      :effect (and
319        (not (inventory-empty ?targetinv))
320        (inventory-holding ?targetinv ?target)
321        (not (object-of-type ?target Hypothetical))
322        (object-of-type ?target Arrow)
323        (not (inventory-holding ?ingredientinv1 ?ingredient1))
324        (inventory-empty ?ingredientinv1)
325        (not (object-of-type ?ingredient1 Stick))
326        (object-of-type ?ingredient1 Hypothetical)
327        (not (inventory-holding ?ingredientinv2 ?ingredient2))
328        (inventory-empty ?ingredientinv2)
329        (not (object-of-type ?ingredient2 Feather))
330        (object-of-type ?ingredient2 Hypothetical)
331      )
332    )
333    (:action craft-beetroot-soup_0
334      :parameters (?t - tile ?station - object ?ingredientinv1 - inventory ?
            ingredient1 - object ?ingredientinv2 - inventory ?ingredient2 -
            object ?targetinv - inventory ?target - object)
335
336      :precondition (and
337        (agent-at ?t)
338        (object-at ?station ?t)
339        (object-of-type ?station WorkStation)
340        (inventory-holding ?ingredientinv1 ?ingredient1)
341        (object-of-type ?ingredient1 Beetroot)
342        (inventory-holding ?ingredientinv2 ?ingredient2)
343        (object-of-type ?ingredient2 Bowl)
344        (inventory-empty ?targetinv)
345        (object-of-type ?target Hypothetical)
346      )
347      :effect (and
348        (not (inventory-empty ?targetinv))
349        (inventory-holding ?targetinv ?target)
350        (not (object-of-type ?target Hypothetical))
351        (object-of-type ?target BeetrootSoup)
352        (not (inventory-holding ?ingredientinv1 ?ingredient1))
353        (inventory-empty ?ingredientinv1)
354        (not (object-of-type ?ingredient1 Beetroot))
355        (object-of-type ?ingredient1 Hypothetical)
356        (not (inventory-holding ?ingredientinv2 ?ingredient2))
357        (inventory-empty ?ingredientinv2)
358        (not (object-of-type ?ingredient2 Bowl))
359        (object-of-type ?ingredient2 Hypothetical)
360      )
361    )
362    (:action craft-paper_0
363      :parameters (?t - tile ?station - object ?ingredientinv1 - inventory ?
            ingredient1 - object ?targetinv - inventory ?target - object)
364
365      :precondition (and
366        (agent-at ?t)
367        (object-at ?station ?t)
368        (object-of-type ?station WorkStation)
369        (inventory-holding ?ingredientinv1 ?ingredient1)
370        (object-of-type ?ingredient1 SugarCane)
371        (inventory-empty ?targetinv)
372        (object-of-type ?target Hypothetical)
373      )
374      :effect (and
375        (not (inventory-empty ?targetinv))
376        (inventory-holding ?targetinv ?target)
377        (not (object-of-type ?target Hypothetical))
378        (object-of-type ?target Paper)
```

```
379      (not (inventory-holding ?ingredientinv1 ?ingredient1))
380      (inventory-empty ?ingredientinv1)
381      (not (object-of-type ?ingredient1 SugarCane))
382      (object-of-type ?ingredient1 Hypothetical)
383    )
384  )
385  (:action craft-shears2_2
386    :parameters (?t - tile ?station - object ?ingredientinv1 - inventory ?
         ingredient1 - object ?targetinv - inventory ?target - object)
387
388    :precondition (and
389      (agent-at ?t)
390      (object-at ?station ?t)
391      (object-of-type ?station WorkStation)
392      (inventory-holding ?ingredientinv1 ?ingredient1)
393      (object-of-type ?ingredient1 GoldIngot)
394      (inventory-empty ?targetinv)
395      (object-of-type ?target Hypothetical)
396    )
397    :effect (and
398      (not (inventory-empty ?targetinv))
399      (inventory-holding ?targetinv ?target)
400      (not (object-of-type ?target Hypothetical))
401      (object-of-type ?target Shears)
402      (not (inventory-holding ?ingredientinv1 ?ingredient1))
403      (inventory-empty ?ingredientinv1)
404      (not (object-of-type ?ingredient1 GoldIngot))
405      (object-of-type ?ingredient1 Hypothetical)
406    )
407  )
408  (:action craft-bowl_1
409    :parameters (?t - tile ?station - object ?ingredientinv1 - inventory ?
         ingredient1 - object ?ingredientinv2 - inventory ?ingredient2 -
         object ?targetinv - inventory ?target - object)
410
411    :precondition (and
412      (agent-at ?t)
413      (object-at ?station ?t)
414      (object-of-type ?station WorkStation)
415      (inventory-holding ?ingredientinv1 ?ingredient1)
416      (object-of-type ?ingredient1 WoodPlank)
417      (inventory-holding ?ingredientinv2 ?ingredient2)
418      (object-of-type ?ingredient2 WoodPlank)
419      (inventory-empty ?targetinv)
420      (object-of-type ?target Hypothetical)
421    )
422    :effect (and
423      (not (inventory-empty ?targetinv))
424      (inventory-holding ?targetinv ?target)
425      (not (object-of-type ?target Hypothetical))
426      (object-of-type ?target Bowl)
427      (not (inventory-holding ?ingredientinv1 ?ingredient1))
428      (inventory-empty ?ingredientinv1)
429      (not (object-of-type ?ingredient1 WoodPlank))
430      (object-of-type ?ingredient1 Hypothetical)
431      (not (inventory-holding ?ingredientinv2 ?ingredient2))
432      (inventory-empty ?ingredientinv2)
433      (not (object-of-type ?ingredient2 WoodPlank))
434      (object-of-type ?ingredient2 Hypothetical)
435    )
436  )
437  (:action craft-boat_0
438    :parameters (?t - tile ?station - object ?ingredientinv - inventory ?
         ingredient - object ?targetinv - inventory ?target - object)
439
```

```
440    : precondition (and
441      (agent−at ?t)
442      (object−at ?station ?t)
443      (object−of−type ?station WorkStation)
444      (inventory−holding ?ingredientinv ?ingredient)
445      (object−of−type ?ingredient WoodPlank)
446      (inventory−empty ?targetinv)
447      (object−of−type ?target Hypothetical)
448    )
449    : effect (and
450      (not (inventory−empty ?targetinv))
451      (inventory−holding ?targetinv ?target)
452      (not (object−of−type ?target Hypothetical))
453      (object−of−type ?target Boat)
454      (not (inventory−holding ?ingredientinv ?ingredient))
455      (inventory−empty ?ingredientinv)
456      (not (object−of−type ?ingredient WoodPlank))
457      (object−of−type ?ingredient Hypothetical)
458    )
459  )
460  (:action craft−cooked−potato_1
461    : parameters (?t − tile ?station − object ?ingredientinv1 − inventory ?
            ingredient1 − object ?ingredientinv2 − inventory ?ingredient2 −
            object ?targetinv − inventory ?target − object)
462
463    : precondition (and
464      (agent−at ?t)
465      (object−at ?station ?t)
466      (object−of−type ?station WorkStation)
467      (inventory−holding ?ingredientinv1 ?ingredient1)
468      (object−of−type ?ingredient1 Potato)
469      (inventory−holding ?ingredientinv2 ?ingredient2)
470      (object−of−type ?ingredient2 Coal)
471      (inventory−empty ?targetinv)
472      (object−of−type ?target Hypothetical)
473    )
474    : effect (and
475      (not (inventory−empty ?targetinv))
476      (inventory−holding ?targetinv ?target)
477      (not (object−of−type ?target Hypothetical))
478      (object−of−type ?target CookedPotato)
479      (not (inventory−holding ?ingredientinv1 ?ingredient1))
480      (inventory−empty ?ingredientinv1)
481      (not (object−of−type ?ingredient1 Potato))
482      (object−of−type ?ingredient1 Hypothetical)
483      (not (inventory−holding ?ingredientinv2 ?ingredient2))
484      (inventory−empty ?ingredientinv2)
485      (not (object−of−type ?ingredient2 Coal))
486      (object−of−type ?ingredient2 Hypothetical)
487    )
488  )
489  (:action craft−gold−ingot_1
490    : parameters (?t − tile ?station − object ?ingredientinv1 − inventory ?
            ingredient1 − object ?ingredientinv2 − inventory ?ingredient2 −
            object ?targetinv − inventory ?target − object)
491
492    : precondition (and
493      (agent−at ?t)
494      (object−at ?station ?t)
495      (object−of−type ?station WorkStation)
496      (inventory−holding ?ingredientinv1 ?ingredient1)
497      (object−of−type ?ingredient1 GoldOre)
498      (inventory−holding ?ingredientinv2 ?ingredient2)
499      (object−of−type ?ingredient2 Coal)
500      (inventory−empty ?targetinv)
```

```
501      (object-of-type ?target Hypothetical)
502    )
503    :effect (and
504      (not (inventory-empty ?targetinv))
505      (inventory-holding ?targetinv ?target)
506      (not (object-of-type ?target Hypothetical))
507      (object-of-type ?target GoldIngot)
508      (not (inventory-holding ?ingredientinv1 ?ingredient1))
509      (inventory-empty ?ingredientinv1)
510      (not (object-of-type ?ingredient1 GoldOre))
511      (object-of-type ?ingredient1 Hypothetical)
512      (not (inventory-holding ?ingredientinv2 ?ingredient2))
513      (inventory-empty ?ingredientinv2)
514      (not (object-of-type ?ingredient2 Coal))
515      (object-of-type ?ingredient2 Hypothetical)
516    )
517  )
518  (:action craft-stick_0
519    :parameters (?t - tile ?station - object ?ingredientinv1 - inventory ?
         ingredient1 - object ?targetinv - inventory ?target - object)
520
521    :precondition (and
522      (agent-at ?t)
523      (object-at ?station ?t)
524      (object-of-type ?station WorkStation)
525      (inventory-holding ?ingredientinv1 ?ingredient1)
526      (object-of-type ?ingredient1 WoodPlank)
527      (inventory-empty ?targetinv)
528      (object-of-type ?target Hypothetical)
529    )
530    :effect (and
531      (not (inventory-empty ?targetinv))
532      (inventory-holding ?targetinv ?target)
533      (not (object-of-type ?target Hypothetical))
534      (object-of-type ?target Stick)
535      (not (inventory-holding ?ingredientinv1 ?ingredient1))
536      (inventory-empty ?ingredientinv1)
537      (not (object-of-type ?ingredient1 WoodPlank))
538      (object-of-type ?ingredient1 Hypothetical)
539    )
540  )
541  (:action craft-sword_0
542    :parameters (?t - tile ?station - object ?ingredientinv1 - inventory ?
         ingredient1 - object ?ingredientinv2 - inventory ?ingredient2 -
         object ?targetinv - inventory ?target - object)
543
544    :precondition (and
545      (agent-at ?t)
546      (object-at ?station ?t)
547      (object-of-type ?station WorkStation)
548      (inventory-holding ?ingredientinv1 ?ingredient1)
549      (object-of-type ?ingredient1 Stick)
550      (inventory-holding ?ingredientinv2 ?ingredient2)
551      (object-of-type ?ingredient2 IronIngot)
552      (inventory-empty ?targetinv)
553      (object-of-type ?target Hypothetical)
554    )
555    :effect (and
556      (not (inventory-empty ?targetinv))
557      (inventory-holding ?targetinv ?target)
558      (not (object-of-type ?target Hypothetical))
559      (object-of-type ?target Sword)
560      (not (inventory-holding ?ingredientinv1 ?ingredient1))
561      (inventory-empty ?ingredientinv1)
562      (not (object-of-type ?ingredient1 Stick))
```

```
563      (object-of-type ?ingredient1 Hypothetical)
564      (not (inventory-holding ?ingredientinv2 ?ingredient2))
565      (inventory-empty ?ingredientinv2)
566      (not (object-of-type ?ingredient2 IronIngot))
567      (object-of-type ?ingredient2 Hypothetical)
568    )
569  )
570  (:action craft-bed_1
571   :parameters (?t - tile ?station - object ?ingredientinv1 - inventory ?
          ingredient1 - object ?ingredientinv2 - inventory ?ingredient2 -
          object ?targetinv - inventory ?target - object)
572
573   :precondition (and
574     (agent-at ?t)
575     (object-at ?station ?t)
576     (object-of-type ?station WorkStation)
577     (inventory-holding ?ingredientinv1 ?ingredient1)
578     (object-of-type ?ingredient1 WoodPlank)
579     (inventory-holding ?ingredientinv2 ?ingredient2)
580     (object-of-type ?ingredient2 Wool)
581     (inventory-empty ?targetinv)
582     (object-of-type ?target Hypothetical)
583   )
584   :effect (and
585     (not (inventory-empty ?targetinv))
586     (inventory-holding ?targetinv ?target)
587     (not (object-of-type ?target Hypothetical))
588     (object-of-type ?target Bed)
589     (not (inventory-holding ?ingredientinv1 ?ingredient1))
590     (inventory-empty ?ingredientinv1)
591     (not (object-of-type ?ingredient1 WoodPlank))
592     (object-of-type ?ingredient1 Hypothetical)
593     (not (inventory-holding ?ingredientinv2 ?ingredient2))
594     (inventory-empty ?ingredientinv2)
595     (not (object-of-type ?ingredient2 Wool))
596     (object-of-type ?ingredient2 Hypothetical)
597   )
598  )
599  (:action craft-iron-ingot_2
600   :parameters (?t - tile ?station - object ?ingredientinv1 - inventory ?
          ingredient1 - object ?ingredientinv2 - inventory ?ingredient2 -
          object ?targetinv - inventory ?target - object)
601
602   :precondition (and
603     (agent-at ?t)
604     (object-at ?station ?t)
605     (object-of-type ?station WorkStation)
606     (inventory-holding ?ingredientinv1 ?ingredient1)
607     (object-of-type ?ingredient1 IronOre)
608     (inventory-holding ?ingredientinv2 ?ingredient2)
609     (object-of-type ?ingredient2 Coal)
610     (inventory-empty ?targetinv)
611     (object-of-type ?target Hypothetical)
612   )
613   :effect (and
614     (not (inventory-empty ?targetinv))
615     (inventory-holding ?targetinv ?target)
616     (not (object-of-type ?target Hypothetical))
617     (object-of-type ?target IronIngot)
618     (not (inventory-holding ?ingredientinv1 ?ingredient1))
619     (inventory-empty ?ingredientinv1)
620     (not (object-of-type ?ingredient1 IronOre))
621     (object-of-type ?ingredient1 Hypothetical)
622     (not (inventory-holding ?ingredientinv2 ?ingredient2))
623     (inventory-empty ?ingredientinv2)
```

```
624      (not (object-of-type ?ingredient2 Coal))
625      (object-of-type ?ingredient2 Hypothetical)
626    )
627   )
628   )
```

**Learned Operator Libraries on ALFRED** The following shows the full PDDL domain definition including the initial provided vocabulary of symbolic environment constants and predicates, initial pick and place operators, and all ensuing learned operators.

```
1    (define (domain alfred)
2        (:requirements :adl
3        )
4        (:types
5            agent location receptacle object rtype otype
6        )
7        (:constants
8            CandleType - otype
9            ShowerGlassType - otype
10           CDType - otype
11           TomatoType - otype
12           MirrorType - otype
13           ScrubBrushType - otype
14           MugType - otype
15           ToasterType - otype
16           PaintingType - otype
17           CellPhoneType - otype
18           LadleType - otype
19           BreadType - otype
20           PotType - otype
21           BookType - otype
22           TennisRacketType - otype
23           ButterKnifeType - otype
24           ShowerDoorType - otype
25           KeyChainType - otype
26           BaseballBatType - otype
27           EggType - otype
28           PenType - otype
29           ForkType - otype
30           VaseType - otype
31           ClothType - otype
32           WindowType - otype
33           PencilType - otype
34           StatueType - otype
35           LightSwitchType - otype
36           WatchType - otype
37           SpatulaType - otype
38           PaperTowelRollType - otype
39           FloorLampType - otype
40           KettleType - otype
41           SoapBottleType - otype
42           BootsType - otype
43           TowelType - otype
44           PillowType - otype
45           AlarmClockType - otype
46           PotatoType - otype
47           ChairType - otype
48           PlungerType - otype
49           SprayBottleType - otype
50           HandTowelType - otype
51           BathtubType - otype
52           RemoteControlType - otype
53           PepperShakerType - otype
54           PlateType - otype
```

```
55            BasketBallType – otype
56            DeskLampType – otype
57            FootstoolType – otype
58            GlassbottleType – otype
59            PaperTowelType – otype
60            CreditCardType – otype
61            PanType – otype
62            ToiletPaperType – otype
63            SaltShakerType – otype
64            PosterType – otype
65            ToiletPaperRollType – otype
66            LettuceType – otype
67            WineBottleType – otype
68            KnifeType – otype
69            LaundryHamperLidType – otype
70            SpoonType – otype
71            TissueBoxType – otype
72            BowlType – otype
73            BoxType – otype
74            SoapBarType – otype
75            HousePlantType – otype
76            NewspaperType – otype
77            CupType – otype
78            DishSpongeType – otype
79            LaptopType – otype
80            TelevisionType – otype
81            StoveKnobType – otype
82            CurtainsType – otype
83            BlindsType – otype
84            TeddyBearType – otype
85            AppleType – otype
86            WateringCanType – otype
87            SinkType – otype
88
89            ArmChairType – rtype
90            BedType – rtype
91            BathtubBasinType – rtype
92            DresserType – rtype
93            SafeType – rtype
94            DiningTableType – rtype
95            SofaType – rtype
96            HandTowelHolderType – rtype
97            StoveBurnerType – rtype
98            CartType – rtype
99            DeskType – rtype
100           CoffeeMachineType – rtype
101           MicrowaveType – rtype
102           ToiletType – rtype
103           CounterTopType – rtype
104           GarbageCanType – rtype
105           CoffeeTableType – rtype
106           CabinetType – rtype
107           SinkBasinType – rtype
108           OttomanType – rtype
109           ToiletPaperHangerType – rtype
110           TowelHolderType – rtype
111           FridgeType – rtype
112           DrawerType – rtype
113           SideTableType – rtype
114           ShelfType – rtype
115           LaundryHamperType – rtype
116
117        )
118        ;; Predicates defined on this domain. Note the types for each
           predicate.
```

```
119  (:predicates
120      (atLocation ?a – agent ?l – location)
121          (receptacleAtLocation ?r – receptacle ?l – location)
122          (objectAtLocation ?o – object ?l – location)
123          (inReceptacle ?o – object ?r – receptacle)
124          (receptacleType ?r – receptacle ?t – rtype)
125          (objectType ?o – object ?t – otype)
126          (holds ?a – agent ?o – object)
127          (holdsAny ?a – agent)
128          (holdsAnyReceptacleObject ?a – agent)
129
130          (openable ?r – receptacle)
131          (opened ?r – receptacle)
132          (isClean ?o – object)
133          (cleanable ?o – object)
134          (isHot ?o – object)
135          (heatable ?o – object)
136          (isCool ?o – object)
137          (coolable ?o – object)
138          (toggleable ?o – object)
139          (isToggled ?o – object)
140          (sliceable ?o – object)
141          (isSliced ?o – object)
142  )
143  (:action PickupObjectNotInReceptacle
144          :parameters (?a – agent ?l – location ?o – object)
145          :precondition (and
146              (atLocation ?a ?l)
147              (objectAtLocation ?o ?l)
148              (not (holdsAny ?a))
149              (forall
150                  (?re – receptacle)
151                  (not (inReceptacle ?o ?re))
152              )
153          )
154          :effect (and
155              (not (objectAtLocation ?o ?l))
156              (holds ?a ?o)
157              (holdsAny ?a)
158          )
159      )
160
161  (:action PutObjectInReceptacle
162          :parameters (?a – agent ?l – location ?ot – otype ?o – object ?r
        – receptacle)
163          :precondition (and
164              (atLocation ?a ?l)
165              (receptacleAtLocation ?r ?l)
166              (objectType ?o ?ot)
167              (holds ?a ?o)
168              (not (holdsAnyReceptacleObject ?a))
169          )
170          :effect (and
171              (inReceptacle ?o ?r)
172              (not (holds ?a ?o))
173              (not (holdsAny ?a))
174              (objectAtLocation ?o ?l)
175          )
176      )
177
178  (:action PickupObjectInReceptacle
179          :parameters (?a – agent ?l – location ?o – object ?r – receptacle
        )
180          :precondition (and
181              (atLocation ?a ?l)
```

```
182             ( objectAtLocation ?o ?l )
183             ( inReceptacle ?o ?r )
184             ( not ( holdsAny ?a ) )
185         )
186         : effect ( and
187             ( not ( objectAtLocation ?o ?l ) )
188             ( not ( inReceptacle ?o ?r ) )
189             ( holds ?a ?o )
190             ( holdsAny ?a )
191         )
192     )
193
194     (: action RinseObject_2
195         : parameters (? toolreceptacle − receptacle ?a − agent ?l −
    location ?o − object )
196
197         : precondition ( and
198         ( receptacleType ?toolreceptacle SinkBasinType )
199         ( atLocation ?a ?l )
200         ( receptacleAtLocation ?toolreceptacle ?l )
201         ( objectAtLocation ?o ?l )
202         ( cleanable ?o )
203         )
204         : effect ( and
205         ( isClean ?o )
206         )
207     )
208
209     (: action TurnOnObject_2
210         : parameters (?a − agent ?l − location ?o − object )
211
212         : precondition ( and
213         ( atLocation ?a ?l )
214         ( objectAtLocation ?o ?l )
215         ( toggleable ?o )
216         )
217         : effect ( and
218         ( isToggled ?o )
219         )
220     )
221
222     (: action CoolObject_0
223         : parameters (? toolreceptacle − receptacle ?a − agent ?l −
    location ?o − object )
224
225         : precondition ( and
226         ( receptacleType ?toolreceptacle FridgeType )
227         ( atLocation ?a ?l )
228         ( receptacleAtLocation ?toolreceptacle ?l )
229         ( holds ?a ?o )
230         )
231         : effect ( and
232         ( isCool ?o )
233         )
234     )
235     (: action SliceObject_1
236         : parameters (? toolobject − object ?a − agent ?l − location ?o −
    object )
237
238         : precondition ( and
239         ( objectType ?toolobject ButterKnifeType )
240         ( atLocation ?a ?l )
241         ( objectAtLocation ?o ?l )
242         ( sliceable ?o )
243         ( holds ?a ?toolobject )
```

```
244            )
245            : effect (and
246            (isSliced ?o)
247            )
248    )
249    (: action SliceObject_0
250            : parameters (? toolobject − object ?a − agent ?l − location ?o −
       object )
251
252            : precondition (and
253            (objectType ?toolobject KnifeType)
254            (atLocation ?a ?l)
255            (objectAtLocation ?o ?l)
256            (sliceable ?o)
257            (holds ?a ?toolobject)
258            )
259            : effect (and
260            (isSliced ?o)
261            )
262    )
263    (: action MicrowaveObject_0
264            : parameters (? toolreceptacle − receptacle ?a − agent ?l −
       location ?o − object)
265
266            : precondition (and
267            (receptacleType ?toolreceptacle MicrowaveType)
268            (atLocation ?a ?l)
269            (receptacleAtLocation ?toolreceptacle ?l)
270            (holds ?a ?o)
271            )
272            : effect (and
273            (isHot ?o)
274            )
275    )
276
277    )
```

