# OpenReview forum: "Learning Grounded Action Abstractions from Language"
_ICLR.cc/2024/Conference — ICLR 2024 poster_

### Official Review · Reviewer_geyi · 2023-10-21

**Soundness:** 3 good
**Presentation:** 3 good
**Contribution:** 2 fair
**Rating:** 6
**Confidence:** 4

**Summary:**

The paper proposes a method that learns a library of symbolic action abstractions (i.e., high-level actions) using LLMs.
Given a natural language instruction and a state (object classes, predicates, etc.), the proposed method uses LLMs to plan a sequence of high-level actions.
In the case of the undefined operator of any high-level action, it further uses LLMs to define the operator based on in-context examples.
Such obtained operators are iteratively refined.
The low-level actions to conduct each high-level action are acquired by BFS such that they satisfy the desired subgoal state.
The low-level action policies are updated based on rewards from the environment after task completion with the policies.
The proposed method outperforms baselines in its empirical validations based on ALFRED and Mini Mincraft.

**Strengths:**

- The paper is generally written well and easy to follow.
- The paper tackles an important issue of action grounding present in LLMs used for action planning.
- Using LLMs to acquire high-level actions and their unknown operators looks reasonable and intriguing.
- The two-staged pipeline for the generation of candidate operator definition is well-motivated and sounds sensible.
- The proposed method achieves strong performance over the baselines by noticeable margins.

**Weaknesses:**

- Some assumptions made are a bit practically unrealistic. For example, $\Phi$ is assumed to be perfect and all environmental information is known, but they are usually not the case, especially for task planning for robotic agents (Shridhar et al., 2020; Krantz et al., 2020; Weihs et al., 2021).
- Obtaining high-level action abstractions needs the corresponding low-level policies to generate low-level actions but it seems that it requires extensive interaction with environments (e.g., brute-force search to find a low-level action sequence that satisfies the subgoal condition). Can the proposed method be also applied to some offline scenarios without these interactive environments? And is there any efficient approach to this?
- The detail of the baseline in Table 1 is unclear. For example, for "Code Policy Prediction," the authors prompt the LLM to predict 1) imperative code policies in Python and 2) the function call sequences with arguments. What is the modification made for each?

\* Krantz et al. Beyond the Nav-Graph: Vision-and-Language Navigation in Continuous Environments. In ECCV, 2020.

\* Weihs et al. Visual Room Rearrangement. In CVPR, 2021.

**Questions:**

See weaknesses above.

---

> ### Author Response · Authors · 2023-11-16
> **Thank you for your review!**
>
> Thank you for your thoughtful review and positive comments on our work! We answer your questions and general clarifications below:
>
> **Are the planning assumptions realistic?**
> The assumptions we make are standard for many grounded instruction following settings in prior work (including many of those that use the ALFRED benchmark, recent work that uses LLMs to interact with the Minecraft domain through the implemented Mineflayer API, like our Voyager baseline.)
> That being said, our approach should actually be applicable to environments under uncertainty. The symbolic operators that we learn do not *need* to assume access to perfectly downward refinable predicates (and in fact, in ALFRED, as we discuss in 4.1, the symbolic predicates are not actually a perfect model of low-level geometric detail in the environment) -- we give this formulation because it helps clarify the optimal learning case. The low-level policy learning technique we use is also a general one that is applicable to arbitrary stochasticity in the underlying environment as well. Thanks, and we will update our paper to clarify this!
>
> **Can the proposed method be also applied to some offline scenarios without these interactive environments? And is there any efficient approach to this?**
> This is also an interesting idea that we can discuss as a direction for future work. Our paper is focused on the online RL learning setting (and uses a standard formulation for online policy learning), but our formulation could be adapted to use offline data in several ways and we can discuss this in future work. For instance, we might easily imagine initializing our low-level policy learning algorithm with direct supervision on offline expert demonstrations and state transitions. However, our approach is primarily geared at the online learning setting because we think it's much more interesting -- if you had a set of offline annotated expert demonstrations, there might be less need to use language as a prior to learn good action abstractions.
>
> **The detail of the baseline in Table 1 is unclear. For example, for "Code Policy Prediction," the authors prompt the LLM to predict 1) imperative code policies in Python and 2) the function call sequences with arguments. What is the modification made for each?**
> Thank you for your feedback -- we will update our paper to try and further clarify this baseline, though as it is intended to reimplement key aspects of the Voyager model, we also refer to that for details. Following that paper (and adapting it to be as close as possible to the learning setting in our work), the code policy prediction model also attempts to learn and refine a library of skills in two stages -- these stages just involve Python code (instead of planner-compatible PDDL operators) and use the LLM as a planner instead of Fast-Downward + Policy search. In stage (1), much like our task decomposition step, we condition on the lingusitic goal and ask the LLM to produce the names and the symbolic code definitions of functions with arguments that define composable policies (eg. for *picking_up_wood* as part of an overall plan to *make an axe*). Then, in (2), as in Voyager, for each specific task, conditioned on the linguistic task description and a full symbolic description of the initial environment state, we allow the LLM to import these previously defined functions (or define new ones on the fly), and produce a plan as a sequence of grounded function calls with their arguments. We execute this sequence of grounded function calls to see if it satisfies the goal to score task accuracy on the benchmark.
>
> Let us know if we can provide further clarification, thank you!

---

> > ### Comment · Reviewer_geyi · 2023-11-22
> > **Official Comment by Reviewer geyi**
> >
> > I thank the authors for their detailed response.
> > The response addressed my concerns raised, so I'd like to keep my positive rating.

---

### Official Review · Reviewer_ikxW · 2023-10-24

**Soundness:** 2 fair
**Presentation:** 4 excellent
**Contribution:** 3 good
**Rating:** 8
**Confidence:** 4

**Summary:**

The authors present a method to iteratively learn a set of action operators that can be used by a symbolic planner to generate high-level plans that are then refined into a series of low-level plans. The action operators are learned using a LLM and their selection is guided by a reward signal. To propose a set of action operators, the LLM decomposes language-based instructions into series of high-level actions. High-level actions that do not have a corresponding action operator are then passed to the LLM (along with a few-shot prompt) for the LLM to generate an operator definition consisting of the list of variables the action operates over, the preconditions that must be met to execute the action, and the effect the action has on the environment. The set of proposed action operators are then evaluated by planning with them to solve environment tasks, and are scored based on how often they are used and how often their use leads to task success. Only action operators with high scores are retained. The authors evaluate on Mini Minecraft and Alfred, and compare against several baselines that use LLMs to provide a low-level sequence of actions, specify subgoals, and to specify the plan as code. Across tasks and baselines, the proposed method performs best.

**Strengths:**

- The paper is very well written and easy to follow.
- The authors incorporate LLM to address the challenging problem of identifying abstract actions.
- The baselines evaluate different ways to incorporate LLMs into planning and action selection tasks.
- The environment on which the methods are evaluated assess actions of different complexity.

**Weaknesses:**

- It would be helpful in the results section, "How does or approach compare to using the LLM to predict just goals, or predict task sequences?", to call out the specific parts of the proposed algorithm that address the limitations observed in the baseline approaches.
- It would have been beneficial to include experiments with multiple LLMs in order to understand the required LLM characteristics.
- It is not clear from the reported experiments and results how much noise the system can handle.
- There are no comparisons to systems that rely on hand-coded action abstractions or other methods for identifying/learning the action abstractions.

**Questions:**

- In section 3.2, the authors state that at each iteration operators are learned for only those tasks that were not solved in the previous iteration. How often were the found plans subpar? For example, taking unnecessary, but valid actions? In the experiment section, the tasks are listed as randomly ordered. How sensitive was the found action operator library to the task ordering?
- In the results section the authors discuss Alfred failure cases as including "operator over specification". When over specification occurred, were multiple instances of the action with different objects seen? For example, a slice object with butter knife and one with steak knife. Were the over specifications arbitrary or driven by the training data? For example, a steak knife was chosen even through a butter knife would also work versus the sharpness level was needed to cut the object.
- The authors suggest that encouraging more diverse proposals could address the failure mode. Was soliciting more diverse proposals attempted? Why would more diverse proposals address operator over specification?
- Why Alfred instead of Habitat?
- How accurate were the different parts of the LLM's output? How correct were the mappings from language description to goal specification?
- Might the Mini Minecraft experiments, while good to test how composable the action abstractions are, simplify the action abstraction process by learning the action abstractions on the simpler tasks for which it is easier to identify action abstractions that are more primitive? Compared to learning the actions on the compositional tasks to see how well the method is able to identify useful and flexibly reusable action abstractions?

---

> ### Author Response · Authors · 2023-11-16
> **Thank you for your review!**
>
> Thank you for the detailed review! To clarify and answer your questions:
>
> **In the results section, "How does or approach compare to using the LLM to predict just goals, or predict task sequences?", can you call out the specific parts of the proposed algorithm that address the limitations observed in the baseline approaches?** The section you name (S4.1) does explain why each of these two baselines underperforms relative to our model. However, in revision we can use the additional space to re-iterate how our approach specifically addresses these ablations. To briefly summarize these points here:
> - *Low-level planning only:* As defined, this baseline has no hierarchical planning at all: it directly attempts to search over the low-level action space towards the LLM-predicted goal. This shows that the problems are, indeed, too long-horizon to make direct low-level search feasible. The fact that our approach predicts operators at all -- decomposing an otherwise infeasible long-horizon goal into higher level components that can be searched for sequentially with associated learnable policies -- is what makes hierarchical planning with 'good' operators better than this approach in general; our performance on these benchmarks shows that we *do* in fact learn 'good' operators.
> - *Subgoal-prediction:* As we point out in the results, the fact that the LLM cannot itself accurately predict the sequence of subgoals showcases the value of the explicit high-level planning algorithm that we are able to leverage by learning planner-compatible operators. In our results, we also provide additional qualititative analysis showing that the LLM struggles to accurately track environment state. Our approach uses a symbolic planner at the high-level that definitionally tracks this state and finds satisficing plans.
>
> **How would results change with different LLMs?** Please see our general response for new results using GPT-4. As we show there, our model does not substantially change with GPT-4. We also re-run the Voyager baseline with GPT-4, as this is the closest baseline to our own work, and the original paper specifically uses GPT-4. We find that it does not close the performance gap between that model and ours.
>
> **How much noise can the system handle?** Please see our general response for results from three different replications on all experiments. Overall, we find that our proposed method consistently outperforms all baselines across replications.
>
> **How do results compare with hand-coded action abstractions?** Please see our general response for a *post-hoc analysis* of the learned operators. The ALFRED benchmark provides hand-coded operators for the dataset. Our model recovers *all* of the ground-truth operators (modulo one slightly underspecified case on the Slice operator that we discuss).
>
> **How often are the plans our system learns suboptimal?** Our high-level planning algorithm is FastDownward, so it searches for optimal plans at the high-level given the operator set and goal. Our low-level planning algorithm is a breadth-first planner that also will not take unnecessary actions. This is a benefit of using structured planners -- we should never take 'valid but unnecessary' actions! We *do* qualitatively find that our other baselines, which use LLMs to *predict* sequences of actions (like the code-policy baseline), do not have this guarantee and can take unnecessary actions. We will highlight this!
>
> **How sensitive is our system to task ordering?** Please see our general response on the randomized replications -- we use random task orderings, and our system learns robustly regardless of task ordering. This is because the algorithm itself is agnostic to task ordering. At each iteration, we always first propose operators in parallel, and evaluate operators on a per-task basis over the entire training dataset. Like other library-learning algorithms (eg. DreamCoder), we effectively recover our own 'curriculum', because our system can solve 'easier' problems first as it encounters them -- if it has already learned the right operators -- and then return to more challenging problems at the next iteration.

---

> > ### Author Response · Authors · 2023-11-16
> >
> > **Why does operator overspecification occur, and is this addressed with more diverse proposals?**
> > Overspecification occurs when the LLM happens to propose two distinct operators rather than a single more general one, and the overspecified operators are both verified and retained because each one of them is applicable to some subset of the training environments (so yes, it is driven by the underlying training environment distribution.) Diversity in proposals would address this: we sample n=4 possible operator definitions from the LLM, but sampling more would increase the probability that we could sample the more general operator from the LLM in the first place. We describe our sampling regime in detail in the appendix A1.1.1 (Symbolic Operator Definition), which simply involves prompting the LLM to produce  distinct operator definitions in sequence (rather than drawing n independent samples) to encourage diversity.
> >
> > We do point out that, as per our post-hoc analysis, the Slice example is actually the *only* learned operator that is overspecified relative to the ground truth library, suggesting that our current operator sampling procedure is already close to optimal for this setting.
> >
> > **Why Alfred instead of Habitat?**
> > Our system can definitely be applied to solving similar tasks in Habitat. We chose Alfred as our evaluation benchmark because Alfred contains more diverse object state change actions such as cooling and heating, while Habitat focuses mostly exclusively on navigation and pick-and-place.
> >
> > **How accurate were the different parts of the LLM's output? How correct were the mappings from language description to goal specification?**
> > Please see our general response for a post-hoc analysis showing the relative accuracy of the goal proposal and operator learning in our pipeline on ALFRED.
> > We do not show Minecraft because our results report 100% accuracy on all benchmarks, but on replications, over three different runs, goal translation is 100% (as this uses synthetic language) and on average the system identifies 12/13 ground-truth crafting operators and 10/10 ground-truth mining operators.
> >
> > **Might the Mini Minecraft experiments, while good to test how composable the action abstractions are, simplify the action abstraction process by learning the action abstractions on the simpler tasks?**
> > We actually see this as a strength of our approach -- natural task distributions in the world have both simpler and more complex tasks (that require learning and reusing skills from simple tasks on harder ones), and our iterative algorithm can automatically build its own curriculum by solving easy tasks, using these tasks to verify skills, and then composing them to harder ones.
> >
> > We do point out that the ALFRED experiments further make this point, as the benchmark itself mixes together both easy and hard tasks in the same task distribution, and our approach is still able to recover a good action decomposition during the iterative learning setting.
> >
> > That being said, comparing to a setting where we directly learn from *only* the most challenging tasks is a good idea for future work. That being said, as we mention in our response to xxKc, the tasks in ALFRED and even the 'simpler' Minecraft tasks are already quite long-horizon in general, so we believe these benchmarks already are a good test of task decomposition.

---

> > > ### Comment · Reviewer_ikxW · 2023-11-21
> > > **Thank your for your answers**
> > >
> > > Thank you for your responses and the additional results.
> > >
> > > As for operator overspecification, occurring only once Alfred is great and the data distribution being the cause makes sense.  It would be great to have a few sentences touching on how this might generalize to the real world.

---

> > > > ### Author Response · Authors · 2023-11-22
> > > > **Response to additional clarification**
> > > >
> > > > Hi Reviewer ikxW,
> > > >
> > > > Glad to hear that the clarifications were helpful.
> > > >
> > > > Regarding generalization to the real world: we see a key contribution of this work *as* overcoming a key hurdle for generalizing long-range planning to the real world, as we show that we can use LLMs to learn operators (and associated low-level policies) that recover the same accuracy as ones that would have previously been hand-designed for the same domain. To implement our approach beyond the virtual environments we use here, we see two important next steps: first, as we discuss in S6 (Discussion and Future Work), we hope to test this on household robotics domains in which the low-level policy learning is over finer-grained manipulation actions designed for an actual navigation and manipulation robotic (eg. the ones in Kitchen Worlds: https://github.com/Learning-and-Intelligent-Systems/kitchen-worlds). Second, as we discuss in our general response, we are highly interested in combining our method with methods like those in Migimatsu and Bohg (2022) that enable learning domain-specific symbolic predicates directly from perceptual data over an environment. We will update the draft to include this clearly spotlit in the final discussion, as obviously, generalization to real world settings is the core goal of all long-horizon planning related work.
> > > >
> > > > Thanks again for the helpful review and discussion -- these are all great points that will substantially improve the manuscript. Please let us know if there's anything else that we can answer and clarify!

---

> > > > > ### Author Response · Authors · 2023-11-22
> > > > > **Final comments before review period closes**
> > > > >
> > > > > Hi Reviewer ikxW,
> > > > >
> > > > > Thank you again for taking the time to review this work -- the feedback in your initial review has resulted in several new results (the replications to address noise, and the GPT-4 evaluation) that have definitely shaped the final draft for the better. With the review period closing shortly, please let us know if there's anything else we can help answer here to finalize or increase your score. Thank you!
> > > > >
> > > > > Best,
> > > > > The Authors

---

> ### Comment · Area_Chair_YFUp · 2023-12-04
> **[Important] Response Required to Authors' Rebuttal**
>
> Dear Reviewer ikxW,
>
> As we progress through the review process for ICLR 2024, I would like to remind you of the importance of the rebuttal phase. The authors have submitted their rebuttals, and it is now imperative for you to engage in this critical aspect of the review process.
>
> Please ensure that you read the authors' responses carefully and provide a thoughtful and constructive follow-up. Your feedback is not only essential for the decision-making process but also invaluable for the authors.
>
> Thank you,
>
> ICLR 2024 Area Chair

---

> > ### Comment · Reviewer_ikxW · 2023-12-04
> > **Response to Author's Rebuttal**
> >
> > With the extra experiments and discussion the authors have addressed my questions and concerns, and I am raising my score accordingly.

---

### Official Review · Reviewer_xxKc · 2023-10-28

**Soundness:** 3 good
**Presentation:** 3 good
**Contribution:** 3 good
**Rating:** 6
**Confidence:** 4

**Summary:**

In this paper, the authors propose to exploit the world knowledge of LLMs for learning action abstractions for hierarchical planning. These action abstractions can then be used to solve long-horizon planning problems, by decomposing a goal into subgoals and solving them using bi-level planning. More specifically, given a task and symbolic state, the authors use LLMs to propose symbolic (high-level) operators and their corresponding definitions (in PDDL) which are then used by a bi-level planner to generate a feasible low-level plan. The useful operators (planning-compatible and grounded) are retained in an operator library (i.e. reusable) and used for subsequent tasks, including those that require the composition of learned operators.

**Strengths:**

1. Overall, the paper is well-motivated, clearly written, and supported by strong empirical evidence.
2. The proposed idea of exploiting knowledge of LLMs for action abstraction is intuitive and effective and would be of interest to the planning and decision-making community.

**Weaknesses:**

It would be nice to have some statistics on the length of the plan sequence (both in terms of high-level and low-level actions), and the number of learned operators, to get an understanding of the task complexity (especially for the compositionality experiments) in Minecraft and ALFRED domains. It is hard to get an idea from just the empirical evidence.

**Questions:**

1. In Sec 3.1 (Symbolic operator definition), is there a process through which you identify and/or discount semantically similar (redundant) operators from being added in the operator library since the LLM generation is not conditioned on it (i.e. the LLM is not aware of the existing operators in the library).

2. In Sec 3.4 (Scoring LLM Operator Proposals), the operators are selected based on their executability in the low-level planning, but the overall goal is not accounted for. Wouldn't this lead to the selection of some operators that are just "feasible" but not "useful"?

Minor Comment:
* Sec 3.4: s/b > \tau_r

---

> ### Author Response · Authors · 2023-11-16
> **Thank you for your review!**
>
> Thanks for the thoughtful review and close read -- these are all great clarification points. To answer your questions and provide clarification on your comments:
>
> **Can you provide statistics on task and plan complexity (high and low-level plans) for your benchmarks?** Great question, and we will update our Appendix to include these numbers:
> - On *ALFRED*, high-level plans compose **5-10** high-level operators, and low-level action trajectories have on average **50** low-level actions. There **over 100** objects that the agent can interact with in each interactive environment. We refer to the ALFRED paper (https://arxiv.org/pdf/1912.01734.pdf) for additional details on this benchmark.
> - On *Mini-Minecraft*: **mining** problems have 2-4 high-level actions and on average 6.9 low-level actions; **crafting** have 4-6 high-level actions and on average 10.24 low-level actions; and **compositional** tasks have 2-26 high-level actions and on average 41.5 low-level actions.
>
>
> **How do we identify and discount semantically similar (redundant) operators from being added in the operator library?**
> Also a good question, and one that helps distinguish our work from other skill learning papers (eg. Voyager). We avoid problems with redundancy in two ways:
> - First, the off-the-shelf high-level planning algorithm we use (FastDownward) is designed not to incur any additional computational cost for redundancy in the operator set. This is one major advantage of using a planning-compatible operator representation, rather than general purpose code -- we can leverage existing planners like this one.
> - Second, however, in general we don't want to include semantically redundant operators for library interpretability. We do this by incrementally growing the library as needed during learning -- we always attempt to plan first with the highest scoring operators, then only try out new operators if planning fails (indicating that we may need a new skill). Wewill update the manuscript to clarify this important detail.
> Empirically, as you can see in our Appendix (A.2, Learned Operators), this means we don't learn redundant operators (and as we discuss in the general response, on ALFRED we in fact manage to just recover the ground truth hand-designed operator library without redundancy).
>
> **How do we avoid learning operators that are merely 'feasible' but not 'useful' if the overall goal success is not included in the operator score?** Also a good and subtle point that we will highlight in our revision. This falls out of how the high-level planner algorithm works: it is an optimizing planner that searches for minimum-description length plans that only include operators which are useful for achieving the goal specification. As a consequence, operators are only used in plans in the first place if they are 'useful' towards the predicted goals; they will only be verified at the low-level and retained in the library if they were both useful *and* achievable in the low-level environment.
>
> *Minor Comment: Sec 3.4: s/b > \tau_r:* Thank you for pointing out the typo. We will fix the typo in the updated version of the paper.
>
> Please let us know if we can provide any additional clarification! Thanks again.

---

> > ### Comment · Reviewer_xxKc · 2023-11-18
> > **Requesting further clarifications**
> >
> > Thank you authors for taking the time to write the detailed clarifications. However, I'm still a bit confused about one of your answers.
> >
> > * **identify and discount semantically similar (redundant) operators**: I don't see how *adding new operators only if planning fails* resolves the redundancy issue since the LLM is not aware of the already learned operators. Can you elaborate on the following examples: (1) if a "place *args" operator is present in the library and the LLM suggests a "put *args" operator which is similar to the former, along with some additional operators which were missing earlier and had led to planning failure. (2) if a "clean *args" operator is present in the library and the LLM proposes a "clean_and_cool *args" operator along with some additional operators ... There should be some strict checks to identify this or resolve it at the source by providing the LLM with the known operators. Of course, the latter feels harder to control. I understand the first point that you made about the design of the high-level planner, nevertheless, it seems this library could grow endlessly in more real-world settings, making it computationally expensive for the planner.
> >
> > I'm satisfied with the other answers.

---

> > > ### Author Response · Authors · 2023-11-18
> > > **Response to identifying redundant operators.**
> > >
> > > Dear Reviewer xxKc,
> > >
> > > Thank you for the clarifications and the detailed examples! Following your suggestions, we would like to break them down into the following cases:
> > >
> > > 1. LLMs proposing identical operators as existing ones. This can be done by syntactically checking the preconditions and the effects of two operators. If two operators only differ in their names, then we can safely remove one of them. We did not implement this check because, in practice, we did not observe such cases. However, we will include such checks in future versions. Thanks for the great suggestion!
> > > 2. LLMs proposing operators that are "similar" to existing operators. There will be two subcases. First, LLMs are proposing an operator for achieving the same goal (e.g., heat something) in different ways (e.g., using stoves vs. microwaves). In such a case, it will be beneficial to keep both operators because they are applicable in different scenarios. Second, LLMs are proposing exactly the same operator, except that it gets part of the precondition/effect "wrong". Such an operator will not be kept in the final library because the low-level motion planner will fail to achieve the specified subgoal of that operator.
> > > 3. LLMs proposing operators that are "combinations" of existing operators. While we did not observe exactly the case such as "clean_and_wash," we did observe the case where LLM proposed operators miss some of the preconditions (e.g., in order to use a tool in Minecraft, the agent should have the tool in its inventory). Equivalently, in such cases, LLMs are proposing operators such as "finding_an_axe_and_then_use_it_to_mine_iron." If the subgoal can not be reached within a given search budget for the low-level planner/policy, then we will reject this new operator. By contrast, if such subgoals can be reached by the low-level planner/policy, we will delete the old operator "mine_iron" because using the new operator will give us a shorter high-level plan. After each iteration, we will delete operators that are never used in any problems (in this case, the original "mine_iron" because the planner always favors shorter plans).
> > >
> > > Thank you again for your consideration and we will update the manuscript with these analyses. Please let us know if you have further questions.

---

> > > > ### Comment · Reviewer_xxKc · 2023-11-19
> > > > **Thank You authors**
> > > >
> > > > Thank you for the clarifications. I believe the overall framework sets a nice tone for future work to develop and play around with similar ideas of using LLM for proposing action abstractions while engaging a bi-level planner for grounding. While I still feel that there should be additional checks in place (postprocessing steps), they could be mere engineering tweaks.
> > > >
> > > > I also agree with other reviewers that the distinction between Voyager and your proposed setup should be made more explicit. I have no further questions for now. Cheers!

---

> > > > > ### Author Response · Authors · 2023-11-20
> > > > > **Thank you**
> > > > >
> > > > > Dear Reviewer xxKc,
> > > > >
> > > > > Thank you again for your suggestions. We will incorporate more discussions on redundant operators and comparisons with Voyager in our revision. Please don't hesitate to let us know if there's anything else we can do to increase your score.
> > > > >
> > > > > Thanks,
> > > > > Authors

---

### Official Review · Reviewer_ukwA · 2023-10-30

**Soundness:** 2 fair
**Presentation:** 2 fair
**Contribution:** 2 fair
**Rating:** 5
**Confidence:** 4

**Summary:**

This paper proposed to leverage LLM to solve long-horizon planning problems by dynamically building a library of symbolic action abstractions and learning a low-level policy to execute the subgoals. They conducted experiments and relevant ablation studies on Mini Minecraft and ALFRED benchmarks to demonstrate the effectiveness of the proposed method.

**Strengths:**

+ The method is technically feasible.
+ The writing is easy to follow.
+ Representing abstract actions with symbols and reasoning them with LLM is an interesting attempt.

**Weaknesses:**

+ **Lack of novelty.** This pipeline is reminiscent of the Voyager model, which also aims to tackle long-horizon tasks via the creation of a dynamic skill library. This limits the contributions of this paper. It is better to highlight the differences between this work and Voyager.

+ **Benchmark is too simple.** It should be noted that the test environment used in the current work (Mini Minecraft) is significantly less complex than the original Minecraft version, resulting in reduced task difficulty. Especially the success rate in Mini Minecraft even reaches 100%.

+ **Concerns about generalization ability.** The proposed method relies heavily on symbolic representations. I'm concerned that it may be difficult to generalize to complex real-world environments, which are often not easily symbolized.

+ **Missing important citations.** [1, 2] are also important methods that leverage LLM as the planner to decompose the long-horizon task into subgoals. I suggest the authors to include some discussion and comparison of such methods.

[1] Describe, explain, plan and select: Interactive planning with large language models enables open-world multi-task agents, https://arxiv.org/abs/2302.01560

[2] Ghost in the Minecraft: Generally Capable Agents for Open-World Environments via Large Language Models with Text-based Knowledge and Memory, https://arxiv.org/abs/2305.17144

**Questions:**

As stated in the Weakness part.

---

> ### Author Response · Authors · 2023-11-16
> **Thank you for your review!**
>
> Thank you for your thoughtful review. We address your comments and questions here:
>
> **How is our model different from Voyager?**
> - First of all, we actually *compare* to Voyager, which is reimplemented as one of our baselines (the 'code policy prediction model', which uses an LLM to propose a library of Python-based policies that it retrieves and composes to solve tasks). We find that our model *dramatically outperforms this model* on both the Mini Minecraft and ALFRED benchmarks. Our discussion includes a failure analysis of this model and highlights several reasons why we outperform the Voyager implementation.
> - Unlike Voyager, our algorithm uses the LLM as a *prior* over the task decompositions and action definitions, but nests the LLM within an outer learning objective designed to *refine and verify the library of skills to learn a compact library tailored to the environment and task distribution*. In contrast, we find empirically (as in the original Voyager paper) that the Voyager objective function tends to learn 100s-*1000s* of highly overfit skills that are locally relevant to specific tasks (eg. a function specific to *mining two pieces of wood*, rather than a more general *mine_wood* function). In practice, we find that this overfitting prevents generalization and efficient planning on more complex tasks, especially ones (like ours) that require adapting to novel environments. We suspect that the success of the original Voyager paper on Minecraft is due in part to the fact that it uses a relatively high-level, human-designed API that also appears in the LLM training data.
> - Unlike Voyager, we also aim to specifically learn *planning-compatible, grounded action representations that support existing hierarchical planners from the robotics literature* for long-horizon planning. We show in our experiments that bridging between LLMs and hierarchical planners allows us to solve tasks and generalize to more complex tasks much more accurately than systems that rely on the LLM itself as a planner (consistent with other literature we cite in our related work whcih documents the difficulty of using LLMs direclty to predict plans).
>
> Our new experiments also show that the gap between the Voyager model and our approach is not closed just by using GPT-4 for code policy prediction instead of GPT3.5.
>
>
> **Are these benchmarks too simple?** While a key goal in future work is to show how this approach can scale to other domains (as we discuss in the general discussion), we would actually push back against the claim that these are very simple benchmarks for testing long-horizon planning from linguistic goals.
> Our baseline results demonstrate that both benchmarks (ALFRED and Mini Minecraft) are *very challenging* for a suite of different LLM-based planning methods, including the Voyager-based code-policy baseline. We see our strong performance on this benchmark as a good sign that -- in the optimal case in which we have access to fully specified linguistic goals -- the model can learn and use highly efficient action abstractions to solve the domain of tasks. As we point out in S4 (Domains), Mini Minecraft tasks are actually quite challenging planning tasks -- the longest crafting tasks in this benchmark compose over 26 *high-level skills* and have an action space of *over 2000 potential actions* at each time step, because we define the benchmark to permit the creation of many potential new objects during action execution.
>
> **How generalizable are methods (like ours) that use symbolic environment predicates in planning?** Please refer to our general response for an answer to this!

---

> ### Author Response · Authors · 2023-11-16
>
> ***Additional citations on planning and LLMs***. We will update our related work section on prior LLM and planning work to cite the papers you mention; thank you for bringing these to our attention! To briefly discuss the contributions of our model in relation to these:
> - [1] This paper considers a somewhat orthogonal technique (self-explanation by an LLM to fix initially proposed plans) for improving an initial LLM-produced subgoal sequence (using a basic method that is similar to the 'subgoal prediction baseline' we use in our work.) We agree that this is an important adjacent contribution to using LLMs for planning, with similar Minecraft and ALFRED-based evaluations, and will cite this work; the self-explanation technique to fix plans is likely one that could be integrated positively into our approach and any of the other baselines we report. This model does not seek to directly learn a composable library of hierarchical actions and does not seek to bridge between LLMs and structured hierarchical planning algorithms, which we see as the core contributions of this work, but does share a broader goal of decomposing long-horizon problems into subgoals.
>
> - [2] This paper also uses an LLM to decompose long-horizon planning tasks (focused on the full Minecraft game) into goal specifications and subgoals, and (similar to Voyager) also uses the LLM to retrieve the full range of plans used in prior successful task executions (including to reference previously learned actions). We will also cite this in prior work. However, as we mention in our discussion of the Voyager model, we see the controlled long-horizon experiments (using our Mini Minecraft domain) and the need to learn under ambiguity (in our ALFRED domain) as highlighting the key contribution we make over these approaches, which use an LLM as an all-purpose policy proposal, decomposition, and retrieval model without the compact library-learning objective we frame in our work: as we find our experiments, we believe that planning-specific hierarchical representations and structured planning algorithms play an important role in accurately and efficiently using learned skills in general; and we believe that we contribute a *general library learning objective for learning a compact, efficient skill library adapted to the planning algorithm at hand*. This prevents learning 1,000s of arbitrary "skills" that may be accurate in the short term but overfit to the distribution of goals, and we believe is a key part of learning compositional, intuitive skills.
>
> We hope this has been helpful -- please let us know if you have any additional questions!

---

> ### Author Response · Authors · 2023-11-22
> **Final questions before review period closes**
>
> Hi Reviewer ukwA,
>
> Thank you again for taking the time to review this work. As the crux of your review seemed to hinge on novelty -- especially in comparison to Voyager, which we see our work as substantially improving over both in theoretical framing and empirical results -- please let us know if there's anything else we can help answer here to finalize or increase your score, with the review period closing shortly. Thank you!
>
> Best,
> The Authors

---

> > ### Comment · Reviewer_ukwA · 2023-11-23
> >
> > I appreciate the response. However, I believe the benchmarks are still simple for me. Thus I decided to raise my rating to (5).

---

> > > ### Author Response · Authors · 2023-11-23
> > > **Response to reviewer ukwA**
> > >
> > > Glad to hear that the rebuttal period has been productive and that the clarifications regarding Voyager were helpful.
> > >
> > > Regarding the benchmarks, as we mention above:
> > > - We find that the benchmarks, and especially the learning setting we use here -- with *no initial expert demonstrations* -- are **empirically challenging** for every other baseline we use here, which we believe replicates many standard and current approaches for using LLMs in planning (in addition to Voyager, we find that even LLMs prompted explicitly to decompose the problems with multiple few-shot examples struggle to do so accurately and effectively.) Both benchmarks are designed to highlight challenges that we believe are key for real-world generalization: large action spaces, long horizons, and many objects present in the environment.
> > > - We also point out that the ALFRED benchmark, which we use here, is a standard and widely used grounded-language benchmark that has been used in much SOTA work on this problem (eg. the recent LLM-Planner paper: https://dki-lab.github.io/LLM-Planner/), to take one of many, which we far outperform using our method.
> > > We do discuss both of these points in *S5.Methods* in the section on domains, but will make sure that this is highlighted.
> > >
> > > Hopefully that's helpful, and thanks again for your time and engagement!

---

> ### Comment · Area_Chair_YFUp · 2023-12-04
> **[Important] Response Required to Authors' Rebuttal**
>
> Dear Reviewer ukwA,
>
> As we progress through the review process for ICLR 2024, I would like to remind you of the importance of the rebuttal phase. The authors have submitted their rebuttals, and it is now imperative for you to engage in this critical aspect of the review process.
>
> Please ensure that you read the authors' responses carefully and provide a thoughtful and constructive follow-up. Your feedback is not only essential for the decision-making process but also invaluable for the authors.
>
> Thank you,
>
> ICLR 2024 Area Chair

---

### Official Review · Reviewer_uRvH · 2023-11-01

**Soundness:** 3 good
**Presentation:** 2 fair
**Contribution:** 2 fair
**Rating:** 6
**Confidence:** 3

**Summary:**

The paper studies the challenge of long-horizon planning. To make this more tractable, the authors leverage hierarchical planning using temporal action abstractions, breaking down intricate tasks into manageable subproblems. The novel contribution is a system that harnesses language to derive symbolic action abstractions and associated learnable low-level policies. By querying large language models (LLMs), the system proposes symbolic action definitions, subsequently integrating these into a hierarchical planning framework for grounding and verification. This approach is framed within a multitask-reinforcement-learning objective, where an agent interacts with an environment to solve tasks described in natural language. The ultimate aim is to construct a library of grounded actions that are both planning-compatible and efficient.

**Strengths:**

The system leverages language to derive symbolic action abstractions, a unique approach to decomposing complex tasks, and subsequently verifies them within a hierarchical planning framework, ensuring the practical applicability of the abstractions,  which was tested on two benchmarks, Mini Minecraft and ALFRED, and outperformed other baseline methods that incorporate language models into planning.

The paper presents a commendable effort in bridging the capabilities of large language models with hierarchical planning, the innovative approach of using language to derive action abstractions is particularly noteworthy.

**Weaknesses:**

- Goal Misspecification: Failures on the ALFRED benchmark often occurred due to goal misspecification, where the LLM did not accurately recover the formal goal predicate, especially when faced with ambiguities in human language.

- Policy Inaccuracy: The learned policies sometimes failed to account for low-level, often geometric details of the environment.

- Operator Overspecification: Some learned operators were too specific, e.g., the learned SliceObject operator specified a particular type of knife, leading to planning failures if that knife type was unavailable.

- Limitations in Hierarchical Planning: The paper acknowledges that it doesn't address some core problems in general hierarchical planning. For instance, it assumes access to symbolic predicates representing the environment state and doesn't tackle finer-grained motor planning. The paper also only considers one representative pre-trained LLM and not others like GPT-4.

**Questions:**

Questions:

- The two-stage prompting strategy involves symbolic task decomposition followed by symbolic operator definition. How does the system ensure that the decomposition is optimal or near-optimal for complex tasks?

- The author mentioned that one of the common failures on the ALFRED benchmark was due to goal misspecification, especially when faced with ambiguities in human language. Could you elaborate on how the system currently handles such ambiguities and if there are plans to improve this aspect?

- The paper demonstrates that action libraries from simpler tasks in Mini Minecraft generalize to more complex tasks. Are there plans to test this generalization capability in more diverse environments or tasks outside of the current benchmarks?

- How scalable is the proposed system? Specifically, if the number of tasks or the complexity of the environment increases significantly, how would the system's performance be affected?

Suggestions:

- Might consider introducing an interactive feedback loop where the system can ask clarifying questions when faced with ambiguous goals or tasks. This could help in refining the task understanding and improve planning accuracy.

---

> ### Author Response · Authors · 2023-11-16
> **Thank you for your review!**
>
> Thank you for your thoughtful review and positive feedback about this work! We answer your questions below:
> **How does the system ensure that the decomposition is optimal or near-optimal for complex tasks?**
> Our operator-learning objective function explicitly optimizes for efficient and accurate planning (which we take as our definition of 'optimality') for each individual task and on the domain as whole. *On any individual task*, we use a hierarchical planning algorithm that searches explicitly for high and low-level plans that are accurate (solve the specified goal) and efficient (minimize the plan length, both in the number of operators used and the number of low-level actions taken overall.) We see one of the benefits of learning symbolic operators as precisely this ability to take advantage of efficient planning algorithms from robotics and decision making literatures. Our baselines demonstrate that the hierarchical planner is important, because ablating any part of the hierarchical planner (or substituting it with an LLM as the planner) dramatically reduces performance. *On the domain as a whole*, our scoring function over operators ensures that we learn operators which consistently support efficient and accurate hierarchical planning across the entire set of tasks, including simple and more complex tasks (like those in both of our benchmarks.) At both the individual task and domain level, our formulation allows us to use the LLM as a *proposal* mechanism for task decomposition and operator definition, but nests this proposal within a larger hierarchical planning loop to optimize for learning useful abstractions and producing optimal plans.
>
>
> **Could you elaborate on how the system currently ambiguities in goal specification?**
> As we discuss in the paper, the human-annotated ALFRED benchmark contains many instances of ambiguity in goal specification: for instance, people may ask to *slice a tomato and put it on a table* when there are at least four possible concrete interpretations of which table is intended in the underlying benchmark (eg. the underlying benchmark distinguishes between the dining table, side table, coffee table, and kitchen counter). Currently, for our implementation and all baselines, we address this by explictly prompting the LLM to list N=4 possible interpretations of a given goal description, then attempting to plan towards all N of these interpretations (all results report the best accuracy out of these N=4 possible attempts). We also find empirically that directly prompting the LLM we use in our paper (GPT3.5) to produce N=4 distinct interpretations if ambiguities exist is more effective than sampling N=4 interpretations from the model posterior. This formulation suggests several ways to address ambiguity, and we will update our paper to address this. First, of course, we can simply take a greater number (N>4) of samples, to address cases where there are more than 4 interpretations (which is the case for many goals in ALFRED that have ambiguity in both the intended object and final goal location). More importantly, sampling explicit formal interpretations of each goal allows us to directly quantify uncertainty over the goal interpretation, as we can quantify how many unique interpretations there are (and their conditional probability given the linguistic goal under the LLM) for a given goal. In the future, as you mention, we could use this uncertainty score in interactive settings to decide when to ask a user for clarification, or to provide multiple possible concrete interpretations directly to a user for them to select the intended one. We will add this in our future work, thanks!
>
> **The paper demonstrates that action libraries from simpler tasks in Mini Minecraft generalize to more complex tasks. Are there plans to test this generalization capability in more diverse environments or tasks outside of the current benchmarks?**
> The ALFRED benchmark also tests generalization amongst simple and complex tasks (the original benchmark includes tasks that compose multiple skills, such as the 'heated, sliced' or 'chilled and chopped' tasks that we mention in the paper). More generally, evaluating the relative generalization and compositionality of the learned operators is certainly a key part of our future work, and we will make sure that this is adequately highlighted as a potential strength of this approach, as the learned operators are designed to be composed/planned over via standard robotics planning algorithms (though, as we mention in the general discussion above and in our paper, the tasks we show here are already quite complex in terms of the actual long-range planning involved, as Mini Minecraft involves sequences of 20+ composed operators).

---

> > ### Author Response · Authors · 2023-11-16
> >
> > **How scalable is the proposed system? Specifically, if the number of tasks or the complexity of the environment increases significantly, how would the system's performance be affected?**
> > Good question. "Performance" here could refer to the impact on the operators we learn (do we still recover accurate, useful operators) and the time our algorithm takes to run at each epoch through the training or testing dataset.
> > For number of tasks, in general, we would expect our algorithm and all of the baselines we report to scale roughly linearly in the number of tasks, simply because we currently evaluate all baselines by iterating through the randomly ordered dataset and planning for each one in turn. If the number of tasks were very large *and the tasks were very diverse* -- that is, there was a large number of latent skills we might wish to learn across the dataset as a whole -- then both our algorithm and the code-policy ("Voyager"-based) baseline would also require more time to test and verify proposed operator definitions (and we discuss this point similarly in terms of the diversity of the original operator proposals and learning speed, in the general discussion about GPT-4). Because our model learns composable operators, however, we would expect our approach to generally be *more scalable* in terms of generalization to more complex tasks, as we already see in our benchmarks, because it can decompose longer-horizon problems using the learned operators and leverage hierarchical planning techniques (like the FastDownward algorithm) to plan efficiently.
> > As we show in our experiments, our approach is much more accurate than other baselines that do not use hierarchical planning at generalization to more complex tasks precisely because it can leverage the learned operators. We would similarly expect it to continue to generalize to more complex tasks that required even longer horizon planning using those learned skills.
> >
> >
> > *Please refer to the general response for our response about goal-translation accuracy, GPT-4, and the scalability of symbolic predicates.* Thank you!

---

> ### Author Response · Authors · 2023-11-22
> **Final check in before review period closes!**
>
> Hi Reviewer uRvH,
>
> Thank you again for your detailed initial review -- we were encouraged by your positive response to the work! The feedback you gave here (eg. to clarify how goal ambiguity is handled, and to encourage comparison to other LLMs) has resulted in several new experiments that will definitely improve the overall work. With the end of the review period closing shortly, we just wanted to check in to see if there was anything else we could help answer here to finalize or increase your score.
>
> Best,
> The Authors

---

> ### Comment · Area_Chair_YFUp · 2023-12-04
> **[Important] Response Required to Authors' Rebuttal**
>
> Dear Reviewer uRvH,
>
> As we progress through the review process for ICLR 2024, I would like to remind you of the importance of the rebuttal phase. The authors have submitted their rebuttals, and it is now imperative for you to engage in this critical aspect of the review process.
>
> Please ensure that you read the authors' responses carefully and provide a thoughtful and constructive follow-up. Your feedback is not only essential for the decision-making process but also invaluable for the authors.
>
> Thank you,
>
> ICLR 2024 Area Chair

---

### Author Response · Authors · 2023-11-16
**General response (1/3)**

We thank the reviewers for their responses. We're glad that they found our paper to be "well-written and easy to follow" (ikxW), to be a "commendable effort in bridging the capabilities of large language models with hierarchical planning" and an "innovative approach" for deriving action abstractions from language (uRvH). Reviewers found that we introduce a "well-motivated" and "intuitive" algorithm (geyi, xxKc) supported by "strong empirical evidence" (xxKc) showing that our model improves over baselines by "noticeable margins" (geyi).

We are also grateful to have received a variety of constructive comments from all reviewers. We address key comments shared across reviewers below. Below, we include new results on:
- Replications of all reported experiments demonstrating robustness to environment noise, task ordering, and LLM sampling;
- New results using GPT-4 in our approach (and the code-policy baseline);
- A post-hoc analysis showing the relative accuracy of each stage in the goal-translation and operator learning piepline. We also use these results to clarify accuracy relative to hand-coded ground truth operators.
- We also include a general response about the *use of symbolic predicates* in planning, in response to clarification questions from multiple reviewers.

All of these will appear in the revised draft. We respond to reviewer comments and questions in the individual responses.

---

> ### Author Response · Authors · 2023-11-16
> **General response (2/3)**
>
> **How replicable are these results / how robust are the results to random task ordering, environment noise, and LLM sampling?**
> We ran and will update the manuscript to show results from n=3 random initialized replications for our model and baselines. Replications test robustness to noise from the random task ordering during learning; environment noise (in ALFRED); and LLM sampling noise. *In all cases, we find that our reported results are robust across all experimental conditions and see minimal differences across replications; we report standard error for all replications.*
>
>
> *Mini Minecraft (n=3 replications)*
> | Model | Mining | Crafting | Compositional |
> | -------- | -------- | -------- |-------- |
> |  Low-level planning only    | 31% (STE = 0.0%)|   9% (STE = 0.0%) |    9% (STE = 0.0% )   |
> | Subgoal prediction    | 33% (STE = 1.6%)|   36% (STE=5.6%) |    6% (STE=1.7%)  |
> | Code policy prediction    | 15% (STE=1.2%) |   39% (STE=3.2%) |    10% (STE=1.7%)  |
> | **Ours**    | 100% (STE=0.0%) |   100% (STE=7.5%) |    100% (STE=4.1%)  |
>
> *ALFRED (n=3 replications)*
> | Model | Original (simple+compositional tasks) |
> | -------- | -------- |
> |Low-level Planning only | 21% (STE=1.0%)|
> |Subgoal prediction | 2% (STE=0.4%) |
> |Code policy prediction | 2% (STE=0.9%) |
> |**Ours** | 79% (STE=0.9%) |
>
> **What would happen if you used GPT-4 in your model? Would it improve the code-policy ("Voyager") baseline?**
> We try running our model (and the Voyager-based model, as the original Voyager paper specifically mentions GPT-4) using GPT-4 to see if this improves results from the Voyager baseline. We find the following results.
> *Mini Minecraft (GPT-4)*
> | Model | Mining | Crafting | Compositional |
> | -------- | -------- | -------- |-------- |
> |Code policy prediction | 12% | 37% | 11% |
> | **Ours** | 100%     | 100%     | 100%    |
>
> *ALFRED (GPT-4)*
> | Model | Original (simple+compositional tasks) |
> | -------- | -------- |
> |Code policy prediction | 11% |
> |**Ours** | 70%  |
>
> We find that:
> - For code policy prediction (the Voyager-based baseline), swapping in GPT-4 moderately improves performance on the ALFRED benchmark and makes essentially no change to performance on the Mini Minecraft benchmarks. Overall, the code policy model (with GPT3.5 or GPT4) performs far worse than our model.
> - For our model, we find that using GPT-4 may slightly improve the robustness of our results on the mini-Minecraft results (we run 3 replications and see perfect performance after learning; in our replications reported above, the Crafting benchmark has a larger STE). On ALFRED, we find that using GPT-4 leads to similar but slightly worse performance than our reported results with GPT 3.5. Upon inspection, we find that this is because (a) GPT-4 is actually slightly less accurate at goal translation on this benchmark, as reported next, and (b) GPT-4 proposes a larger and more diverse set of initial operators, which takes more time to verify and leads to lower lifelong performance over 2 iterations of learning. We also see this 'greater diversity' in goal translation and operator proposal on minecraft. For learning, this diversity in operator proposals means our approach would additional iterations of learning (by the end of n=2 iterations, we do in fact recover all desirable skills on ALFRED). To address inaccuracy in goal translation, however, we suggest (as we mention in our paper) that in future work, rather than using general-purposed closed-source LLMs, we might instead seek to use LLMs specifically trained on domain-agnostic goal->PDDL translation for this part of the pipeline.

---

> ### Author Response · Authors · 2023-11-16
> **General response (3/3)**
>
> **How accurate is each part of our model pipeline? (goal translation vs. operator learning)**
> This is a good question, especially because our paper is primarily about learning libraries of grounded actions, not about parsing languge into formal goal specifications (which has been addressed in many other LLM+planning works). In our paper, we provide a qualitative failure analysis and mention that goal misspecification contributes to goal accuracy. We ran an additional post-hoc analysis to determine how each part of the pipeline (using LLMs to translate goals into formal specs; versus learning operators themselves) contributes to overall planning accuracy. We will report the following quantitative results for our model on *ALFRED* (as our model achieves 100% accuracy on the Mini Minecraft baselines). We also report these for the GPT-4 run to clarify the results we find above.
>
> *Goal translation on ALFRED:*
> - Our model (GPT 3.5): 92% of the time, the ground truth goal appears as one of the top-4 goals translated by the LLM.
> - Our model (GPT-4): 82% of the time, the ground truth goal appears as one of the top-4 goals translated by the LLM.
>
> *Operator accuracy on ALFRED:*
> To quantify operator accuracy, we manually compare the semantics of the learned operators with the ground-truth set of operators defined by the original ALFRED engineers.
> - Our model (GPT 3.5): the ground-truth operator set contains 8 distinct operators corresponding to different skills. **We learn semantically-identical (eg. same predicate preconditions and postconditions) for *all* of these ground-truth operator**s except for the *Slice* skill, which, as we discuss in the paper, is learned as two separate slicing operators, each using one of the possible 'knife' types to slice. See **Appendix (A.2, Learned Operator Libraries)** for all of these operators.
> - Our model (GPT-4): this model also learns all operators in the ground truth set except for the Slice definition (and we do notice that, unlike GPT3.5, it actually only learns one operator using one of the two possible knife types to slice). As we also discuss above, however, we find that GPT-4 proposes a larger and more diverse initial set of operator definitions than GPT 3.5, and learning therefore takes longer to try, verify, and ultimately refine down the library to this operator set.
>
> **How well would the results compare to a system that uses hand-coded operators designed by a human engineer?**
> On Mini-Minecraft, our model performs at ceiling (100% accuracy) after two iterations of learning.
> On ALFRED, the results above show that we also **recover all of the ground-truth operators** designed by the original ALFRED dataset engineers after learning (except for the fact that we learn two separate Slice operators instead of one).
>
>
> **How generalizable are methods (like ours) that use symbolic environment predicates in planning?**
> The general scalability of symbolic planning operators is an important point that we will extend in our discussion. Our model *does* assume access to an initial set of symbolic predicates that describe the dynamics of an environment. This is a standard assumption in many robotics settings, and indeed, symbolic predicates (for high-level planning) that ground out into lower-level classifiers are a major part of many state-of-the-art Task and Motion Planning systems, motivating our formulation in this paper. Indeed, many of the other papers we cite that integrate LLMs and planners -- including the models we adapt for all of our baselines, and the Voyager model -- make this assumption in order to integrate symbolic planning and goal verification methods or to learn and execute code.
>
> In terms of future scaling, however, we are especially optimistic about the prospects for integrating our algorithm with work that *learns symbolic predicates from raw visual inputs and interactive experience* (eg. Migimatsu and Bohg, 2022), suggesting a route for scaling our approach to operate directly over initial inputs even in settings with no initial predicate classifiers. We will update our paper to cite this line of work as a route towards scaling this approach to learn directly from perceptal experience.
>
>
> Migimatsu and Bohg, Grounding Predicates through Actions, IEEE International Conference on Robotics and Automation (ICRA), 2022

---

### Author Response · Authors · 2023-11-20
**Happy to answer any further questions**

Dear Reviewers,

Thank you for reviewing our submission. We have posted our response per your suggestions and questions. We are happy to discuss with you and answer any further questions. As the deadline for discussion is approaching, we very much look forward to your feedback.

Thanks
Authors

---

### Author Response · Authors · 2023-11-22
**Looking forward to your feedback and discussion**

We would like to thank all reviewers again for their insightful reviews and suggestions. We have posted individual responses to you, and in our general response, we have summarized our new results based on reviewers' suggestions, including:

- replication of all experiments and reported variances.
- new results using GPT-4 as a more powerful LLM.
- break-down analysis of accuracies of different modules

In addition to the general questions answered in our general response, the most significant critique (from reviewer ukwA and the other reviewer who echoed it) was to ensure that this work was distinguished properly from the Voyager paper.
This is important feedback, and while we discuss the differences between voyager and our model explicitly in our current discussion (S5. Results, *How does our approach compare to using the LLM to learn and predict plans using imperative code libraries?*), we will edit the results to further highlight the response shaped by this rebuttal period. While Voyager approaches the related high-level problem of learning libraries of action skills using priors in language, we believe many aspects of this framing are not specific to Voyager, and more importantly, we ***believe our model significantly improves over Voyager, both empirically and conceptually***.

To summarize our points below for the reviewers and the AC:
- We **compare against a Voyager re-implementation (Code Policy Prediction baseline) in our paper, and find that our model dramatically outperforms this model on all of our benchmarks**. In response to reviewer feedback, we also re-run this baseline and our model using GPT-4, and we again find that we still dramatically outperform this model.
- We include a qualitative analysis in *S6. How does our approach compare to using the LLM to learn and predict plans using imperative code libraries?* explaining the empirical difference, which also highlights the key conceptual improvement over Voyager. We will further extend this section based on the rebuttal discussion, which has been helpful: **unlike Voyager, which learns a large number of composable 'code policies' defined in Python over a high-level API, and uses an LLM to chain these in sequence**, we explicitly learn planner-compatible operators** (and corresponding low-level policies) that can be searched over using hierarchical planners, enabling much better generalization to unseen goals.
- We also frame our learning objective specifically around learning a compact library of skills that enables efficient planning, allowing us to learn a **theoretically and empirically more precise and generalizable set of skills** (which almost completely rediscovers those hand-engineered for our benchmarks!), unlike the hundreds of redundant skills that our Voyager implementation learns on the same tasks.

Finally, we thank all reviewers again. As the deadline for discussion is approaching, we very much look forward to your feedback.

---

### Meta-Review · Area_Chair_YFUp · 2023-12-09

**Metareview:**

This paper presents an innovative approach that integrates language models with hierarchical planning to derive symbolic action abstractions, demonstrating its effectiveness on the Mini Minecraft and ALFRED benchmarks. With an average score of 6.2 and detailed, comprehensive responses from the authors, the recommendation is inclined towards acceptance. The strengths of the paper are evident in its unique approach to leveraging language for task decomposition and verifying the practical applicability of abstractions, which outperformed other baseline methods. The authors have addressed key concerns raised by the reviewers, including goal misspecification, policy inaccuracy, operator overspecification, and limitations in hierarchical planning.

**Justification For Why Not Higher Score:**

The recommendation for a poster presentation is due to concerns about the system's goal misspecification and policy inaccuracy, particularly in complex language scenarios and real-world applicability. The limitations in hierarchical planning and challenges in generalizing to more diverse environments contribute to the paper not achieving a higher score.

**Justification For Why Not Lower Score:**

Considering the paper's innovative approach, strong empirical results, and the authors' detailed responses to critical feedback, it is recommended for acceptance. The paper makes a contribution to the fields of hierarchical planning and language model application, bridging a gap between complex task understanding and practical implementation. The authors have demonstrated a commendable effort in enhancing the capabilities of language models for practical and effective task decomposition, which marks an advancement in the field.

---

### Decision · Program_Chairs · 2024-01-16

Accept (poster)